# 11Plus-Bench: Demystifying Multimodal LLM Spatial Reasoning with Cognitive-Inspired Analysis

## Abstract

For human cognitive process, spatial reasoning and perception are closely entangled, yet the nature of this interplay remains underexplored in the evaluation of multimodal large language models (MLLMs). While recent MLLM advancements show impressive performance on reasoning, their capacity for human-like spatial cognition remains an open question. In this work, we introduce a systematic evaluation framework to assess the spatial reasoning abilities of state-of-the-art MLLMs relative to human performance. Central to our work is 11Plus-Bench, a high-quality benchmark derived from realistic standardized spatial aptitude tests. 11Plus-Bench also features fine-grained human expert annotations of both perceptual complexity and reasoning process. These annotations allow us to move beyond aggregated accuracy, enabling an *instance-level, parallel analysis* of human and machine cognitive profiles with *predictive power*. Through extensive experiments across 14 MLLMs and human evaluation, we find that current MLLMs exhibit early signs of spatial cognition. We find both convergence and divergence: while both human and MLLM cognitive effort, measured by response time and tokens generated respectively, correlates with reasoning complexity, their underlying mechanisms differ. Human correctness is highly predictable and shaped by abstract pattern complexity, whereas instance-level MLLM performance remains weakly predictable and sensitive to low-level perceptual features. Our work provides a precise characterization of the emerging yet brittle spatial reasoning in MLLMs, offering actionable insights for developing more human-like spatial intelligence.

## 1 Introduction

Many achievements of Large Language Models (LLMs) (Brown et al., 2020; Ouyang et al., 2022; Anil et al., 2023) and their multimodal variants (MLLMs) (Hurst et al., 2024; Reid et al., 2024; Gemini, 2025) are largely concentrated in domains where reasoning can be framed through symbolic sequence processing, including code generation (Austin et al., 2021; Lai et al., 2023), mathematical problem solving (Lu et al., 2024; Wang et al., 2024b; 2025), and question answering (Yang et al., 2015; Hendrycks et al., 2020; Li et al., 2025b; Zhang et al.). Human intelligence goes beyond symbolic processing. It relies heavily on perceptual intuition and mental imagery to simulate hypothetical scenarios via object-based imagery (e.g., of shapes) and spatial imagery (e.g., of locations) (Moulton & Kosslyn, 2009; Kozhevnikov et al., 2005), which is still underexplored with MLLMs (Li et al., 2025a; Xu et al., 2025b). Spatial reasoning, also referred to spatial intelligence in cognitive science, encompasses all thinking about spatial content: object shape or location, and manipulating, imagining, or inferring relationships between objects in space (Newcombe, 2024). Carroll's Three-Stratum Theory of Intelligence (Carroll, 1993; 1997) places *Visualization* and *Spatial Relations* as core narrow abilities within the general spatial intelligence domain (Gv), contributing to general intelligence (*g*) as evidenced by empirical research (Deary et al., 2010). Spatial reasoning is crucial for success in STEM fields, visuospatial memory, navigation, and mechanical reasoning (Harvey, 1985; Wai et al., 2009; Harris, 2021; Li et al., 2024b; Zhou et al., 2024). Despite its fundamental importance to human intelligence, spatial reasoning remains a relatively underexplored area in the evaluation of artificial intelligence.

Figure 1: **Overview of evaluation framework with 11PLUS-BENCH**, including fine-grained annotations of cognitive features across diverse tasks targeting three core spatial capabilities. These annotations enable predictive modeling of correctness for both humans and MLLMs, followed by cognitive profile analysis to identify key features that influence accuracy and cognitive load.

Existing work evaluating MLLM spatial reasoning has largely relied on aggregate metrics such as overall or task-wise accuracy (Ramakrishnan et al., 2025; Stogiannidis et al., 2025; Xu et al., 2025a), which offers only a coarse view of model ability. These holistic evaluations often conflate distinct cognitive processes, such as perception, symbolic reasoning, and spatial inference (Zhou et al., 2025), limiting interpretability and obscuring a model's true capabilities in spatial reasoning. Consequently, pinpointing specific skill deficits in current systems from aggregated metrics is challenging, leading to potential misattributions (e.g., mistaking perceptual failures for reasoning deficits (Chollet et al., 2024; 2025)) and hindering clear improvement pathways for MLLM spatial cognition. Furthermore, despite referencing human cognitive tests as testbed, comparisons between human cognition and model behavior in existing work remain relatively shallow (Xu et al., 2025a; Wei et al., 2025; Zhang et al., 2025), failing to specifically highlight current MLLM systems' deficiencies compared to human capabilities.

To address these gaps, we ask: Do current MLLMs engage in spatial reasoning in a manner aligned with human cognition? We refer to the strategies and capabilities of perception, interpretation, and reasoning in spatial contexts as the model's cognitive profile, and we aim to facilitate a parallel comparison of these cognitive profiles between humans and MLLMs.

To this end, we present this evaluation framework centered around 11PLUS-BENCH, a newly-introduced high-quality benchmark grounded in standardized spatial aptitude tests used in human cognitive assessments (Uttal et al., 2013; Hodgkiss et al., 2018a). This design isolates spatial reasoning from confounding factors such as commonsense knowledge or numerical ability. Unlike traditional benchmarks that emphasize aggregate accuracy, 11PLUS-BENCH supports **instance-level parallel comparisons with predictive power** between the correctness of model responses and the perceived difficulty of human behaviors. This is achieved through *fine-grained expert annotations* of *cognitive features*, capturing both visual pattern complexity (perceptual load) and reasoning process (inference difficulty). These fine-grained annotations allow us to estimate the likelihood of a correct response for either system (human or models) and to identify the impactful factors influencing the system behaviors. To compare with human performance, we conduct human evaluations with three participants and use response time as a proxy for cognitive load (Barrouillet et al., 2007; Kyllonen & Zu, 2016). Our annotations exhibit high inter-annotator agreement and strong predictive power for participant response time with annotated cognitive features, validating the benchmark's quality

and interpretability. 11PLUS-BENCH also minimizes contamination concerns by collecting expert annotations for data with no golden answers (over 50%) and holding out a test split composed of problems sourced from commercial test providers that are not publicly available.

Experimental results across 14 state-of-the-art MLLMs reveal a substantial performance gap between models and humans, emphasizing the current limitations of MLLMs in spatial reasoning. While advanced proprietary MLLMs show early signs of spatial reasoning ability, their instance-level performance remains random and poorly predictable with human-inspired cognitive features above. Further analysis of human and models uncovers both convergence and divergence in cognitive profiles. Reasoning-related complexity correlates strongly with cognitive load, measured by response time in humans and token counts of response for MLLMs as approximations for test-time computational effort of both systems. However, model performance is more sensitive to understanding low-level visual cues such as image resolution and spatial relations, whereas human accuracy is primarily influenced by abstract pattern complexity. This blend of similarity and divergence reveals both the emergence of spatial reasoning capabilities in MLLMs and their current deficiencies. Unlike humans, whose spatial reasoning is structured, MLLMs often lack the robustness and compositional understanding necessary for consistent, human-like spatial cognition.

## 2 RELATED WORK

**Spatial Aptitude Test in Cognitive Science** Human spatial ability includes *intrinsic* object-centred skills (e.g., mental rotation, paper-folding) and *extrinsic* environment-centred skills (e.g., perspective taking, navigation) (Hodgkiss et al., 2018b). Classic experimental work on mental rotation by Shepard & Metzler (1971) and Cooper (1975) frames rotation as a continuous internal transformation. Factor-analytic syntheses later showed that rotation loads on a separable spatial factor distinct from verbal or numerical reasoning (McGee, 1979; Linn & Petersen, 1985; Carroll, 1993). Perspective-taking studies, notably Hegarty & Waller (2004), demonstrated a double dissociation from mental rotation, motivating multi-dimensional test batteries such as the Vandenberg–Kuse Mental Rotation Test, Paper-Folding and Spatial Orientation tests (Ekstrom & Harman, 1976). Training meta-analyses confirm that spatial skills are malleable and transfer to STEM success (Cheng & Mix, 2014; Uttal et al., 2013). Neuropsychological reviews link these competencies to parietal–frontal circuits and hippocampal place/grid coding, underscoring their foundational role in cognition (Burgess, 2008; Husain & Nachev, 2007). Together, these findings provide both theoretical structure and validated psychometrics that any AI-oriented spatial benchmark should respect.

**Spatial Cognition with MLLMs** Early multimodal benchmarks such as CLEVR (Johnson et al., 2017b) and NLVR 2 (Suhr et al., 2019) introduced synthetic and natural-image tasks that hinge on recognising static binary relations (e.g., *left of, behind*). Subsequent datasets, e.g. SpatialSense (Yang et al., 2019), Spatial-MM (Shiri et al., 2024), and Comsa & Narayanan's preposition suite (Comsa & Narayanan, 2023), tightened the focus on fine-grained relational semantics. Yet performance plateaus suggest that current MLLMs still rely on language priors rather than genuine geometric reasoning (Wang et al., 2024a; Xu et al., 2025b). Dynamic extensions (CLEVRER (Yi et al., 2020), TopViewRS (Li et al., 2024a), VSI-Bench (Yang et al., 2024)) add temporal sequences, but typically restrict transformations to planar translation or simple collisions, leaving rotation, reflection, and multi-step composite reasoning under-explored. Holistic test batteries such as *MindtheGap* (Stogiannidis et al., 2025), VisFactor (Huang et al., 2025) and SAT (Ray & others, 2024) broaden the coverage by emulating psychometric tasks. Despite the breadth, analyses remain largely descriptive, reporting that "MLLMs fail" without isolating *why* (e.g., frame-of-reference confusion, object-correspondence errors) or benchmarking against human baselines (Ramakrishnan et al., 2025). Our benchmark, 11PLUS-BENCH, adopts a cognitive science–informed taxonomy and includes human performance statistics for each item, enabling detailed, parallel analysis of model and human cognitive profiles.

## 3 11PLUS-BENCH BENCHMARK

### 3.1 COLLECTION OF TASKS

**Spatial Capabilities.** Human cognitive development involves several key capabilities that collectively form spatial intelligence. Psychometric research has identified and quantified these through stan-

dardized tests, capturing dimensions such as Spatial Relation and Orientation, Spatial Visualization, Flexibility of Closure, Perceptual Speed, Spatial Memory, and more (Shepard & Metzler, 1971; Linn & Petersen, 1985; Burgess, 2008; Yılmaz, 2009; Johnson et al., 2017a; Wei et al., 2020; Ekstrom & Hill, 2023).

However, not all these capabilities are equally relevant for evaluating current MLLMs, given fundamental differences in reasoning mechanisms between human cognition and machine learning models. For instance, perceptual speed is less critical for current MLLM paradigms, which do not process information in real-time like humans. Similarly, factors like spatial memory (Burgess, 2008; Ekstrom & Hill, 2023) (e.g., recalling routes or locations over time) or kinesthetic spatial reasoning (understanding space through bodily movement) (Presson et al., 1987; Proske & Gandevia, 2009) may not directly translate to current MLLM architectures which primarily operate on simulated static multimodal inputs. Therefore, we select three representative spatial capabilities:

- *Spatial Relation and Orientation (**SRO**)*: Involves understanding relationships between objects in space, including distance, direction, and position (Newcombe & Learmonth, 2005; Yılmaz, 2009). It is essential for tasks requiring recognition of spatial configurations and interrelations.

- *Spatial Visualization* (**SV**): Refers to the ability to mentally manipulate and transform spatial information (Michael et al., 1957; Shepard & Metzler, 1971). This is important for tasks involving mental rotation, pattern recognition, and imagining as well as manipulating objects or scenes.

- *Flexibility of Closure* (**FoC**): Pertains to the ability to perceive and mentally complete incomplete patterns or shapes (Yılmaz, 2009). This cognitive ability is crucial for solving problems that require identification of missing or occluded elements.

**Task Selection.** We utilize well-established psychometric tests corresponding to the selected capabilities (Harris et al., 2013; Lovett & Forbus, 2013; Jirout & Newcombe, 2015; Parkinson & Cutts, 2018; Gunalp et al., 2019; Uttal et al., 2024). These tests are widely acknowledged and developed in cognitive science, ensuring a fair and parallel comparison between AI systems and humans. Because most psychometric tests use diagrams and structured questions as multimodal input, they also allow for controlled experiments in terms of task complexity while controlling other irrelevant factors to spatial intelligence, such as entity recognition in real-world images. Table 1 presents the correspondence between tasks and capabilities, and Figure 1 provides concrete examples. See Appendix B for detailed definitions of each task.

## 3.2 COLLECTION OF COGNITIVE FEATURES

Answering spatial cognition questions not only requires spatial reasoning but also depends on visual perception and general reasoning performance. These factors influence the probability of a correct response from both humans and machines but do not directly measure spatial reasoning. For a fine-grained explainable investigation, we collect performance-relevant *cognitive features* as follows:

**Visual Perception.** More complex patterns require greater cognitive load for humans to perceive and analyze. For both the question and options, we quantify pattern complexity as the number of atomic components in the patterns as key features, defined by how humans perceive and analyze patterns (details on the objective definition of 'key features' can be found in Appendix B).

**General Reasoning.** Longer reasoning chains indicate greater question complexity and a higher likelihood of error (Garey & Johnson, 2002; Johnson-Laird, 2010). Transitions among reasoning types, such as logical deduction and pattern recognition, add extra cognitive load. These features are distinct from intrinsic spatial cognition but influence reasoning time or response correctness. Variations in these features are subjectively profound, as different individuals may adopt different reasoning chains, especially for more complex questions. To account for this subjective variation, we annotate the general reasoning process by requiring human annotators to choose from four predefined categories of atomic operations: *Pattern Matching*, *Spatial Relation Analysis*, *Spatial Manipulation*, and *Logical Deduction*, each comprising a set of specific operations with details in Appendix B.

In addition to these cognitive features, *bounding boxes* of question and option patterns are also collected in pixel coordinates.

## 3.3 BENCHMARK CONSTRUCTION

To facilitate the evaluation framework, we construct the 11PLUS-BENCH with realistic cognitive science test targeted for teenagers aged 11 or above (11PLUS). We keep the original multiple-choice format used in real-world spatial aptitude tests, which enables more straightforward and accurate correctness-based evaluation.

We compile the public portion of our benchmark by crawling the web using 29 carefully chosen spatial reasoning keywords, resulting in 5352 webpages and 29 hours of videos. A rule-based filtering pipeline is then introduced to discard irrelevant, ambiguous, or non-spatial reasoning samples, ensuring data quality and relevance. Implementation details are provided in Appendix B. Concurrently, the private portion of our benchmark is sourced by purchasing materials from official test centers. This dual approach, combining newly annotated public data with proprietary test-center materials, creates a robust and professional dataset that captures a broad spectrum of spatial cognition challenges while ensuring data quality and contamination control for model evaluation.

All annotations were performed by three human experts, who are postgraduate-level or higher with mathematical or engineering backgrounds. Annotators were trained using standardized guidelines to ensure consistency and reliability across the dataset. To ensure data quality, anotators manually filtered the large raw pool down to 824 items that meet strict criteria: no ambiguity, well-defined spatial cognition problems, image quality, as well as potential copyright and privacy issues. They annotated the filtered public set and an additional 100 samples drawn from the private set, creating a diverse and robust foundation for evaluating spatial reasoning. Data examples deemed low-quality, without a correct answer, or not belonging to spatial cognition were manually filtered and discarded. By combining thorough filtering with expert human annotation, we ensure the benchmark reflects genuine spatial cognition challenges and minimizes errors.

**Benchmark Quality Analysis** The fine-grained annotated benchmark contains 824 data points in the public set and 91 data points in the private set after filtering, all annotated by 3 domain experts. The annotations exhibit strong internal consistency and correctness, underscoring the high quality of the dataset, as shown in Figure 2. The annotated answers achieve 94.5% accuracy on private set against gold-standard labels. For subjective fields such as Reasoning Steps, we observe a high level of annotator agreement, with Pearson correlation coefficients typically around or above 0.8. We also observe subjective variation, as indicated by the mean ± variance and exact-match rates for reasoning-step length (4.66 ± 1.15; 39.78%) and fine-grained operation types (4.54 ± 0.47; 50.48%). This indicates that experts converge on the overall cognitive structure for reasoning while retaining individual interpretations with a certain level of inherent subjectivity. The objective pattern complexity for both questions and options shows perfect agreement among annotators, with numbers strictly aligned. Appendix B provides more information about our benchmark.

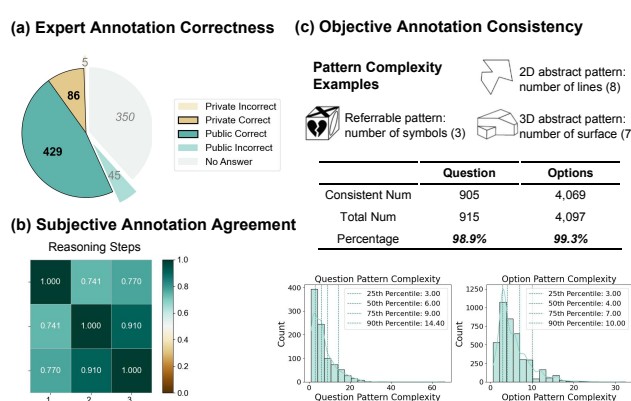

Figure 2: **Quality analysis of expert data collection.** Expert annotations achieve high accuracy on private data with golden answers and exhibit strong agreement across both subjective and objective annotation fields.

**Data Highlights** Here are the key highlights of 11PLUS-BENCH:

- *More Realistic Data*: 11PLUS-BENCH contains two separate data splits (public with 824 examples & private with 91 examples), all derived from realistic 11Plus spatial aptitude test. The public set was crawled from the web, while the private set was purchased from test centers and involves copyrights and intellectual properties.

- *Lower Risk of Data Contamination*: With experts annotating over 50% data with no golden answer available and withholding the private set due to intellectual property considerations, 11PLUS-BENCH significantly lowers the risk of data contamination when evaluating model performance.

- *Richer Cognitive Features*: In addition to the golden answer, 11PLUS-BENCH provides richer fields including not only *bounding boxes* for patterns but also *visual perception* complexity, *general reasoning* process as cognitive annotation.

## 4 EXPERIMENTS AND RESULTS

### 4.1 EXPERIMENTAL SETUPS

**Models**   To comprehensively assess the spatial cognition capabilities of contemporary Multimodal Large Language Models (MLLMs), we selected a diverse suite of 14 models. This selection encompasses both open-sourced and close-sourced architectures, varying significantly in their parameter counts and underlying designs. Specifically, we evaluated four open-sourced models: Qwen-VL-2.5 (Bai et al., 2025) (with 3B and 7B parameters) and Gemma 3 (Team et al., 2025) (with 12B and 27B parameters). Complementing these, we included ten close-sourced MLLMs: GPT-4o, GPT 4.1 mini, GPT 4.1 nano, GPT-o1, GPT-o3, GPT-o4-mini[1], GPT4.1, Gemini 2.0 Flash preview, Gemini 2.5 Flash preview and Gemini 2.5 Pro preview (Hurst et al., 2024; OpenAI, 2024; 2025a;b; Reid et al., 2024; Gemini, 2025). This curated set allows for a broad analysis of how model scale and accessibility correlate with performance.

**Task Settings**   The evaluation methodology extends traditional Visual Question Answering (VQA) benchmarks by also presenting multiple images as options in response to a given question. We investigate two distinct presentation formats to evaluate the MLLMs' spatial cognition:

1. **Single Composite Image**: In this setup, a single image is presented to the model, as with humans. This image contains both the primary image relevant to the question and all candidate option images arranged spatially. This approach is adopted by previous works in benchmarking the spatial cognition performance of MLLMs (Stogiannidis et al., 2025; Ramakrishnan et al., 2025; Xu et al., 2025a).

2. **Separate Images with Bounding Box Annotations**: The primary image and each option image are cropped from the original images as distinct, separate visual inputs. This allows models to potentially ground their reasoning more precisely on specific visual elements.

The performance of the MLLMs across all tasks is quantified by their accuracy in selecting the correct option image that answers the posed question.

**Human Evaluation**   Three participants who are not involved in the annotation process are recruited in order to assess human performance on 11PLUS-BENCH benchmark, strictly adhering to ethical regulations. The examples for human evaluation are uniformly sampled from different tasks, with all data being used for specific task if the available examples are less than sampling requirements, resulting in 402 examples in total. In addition to collecting participants' selected answers, we record the *response time* for each human participant to answer the question, measured in seconds, as an outcome-driven proxy for overall cognitive load (Barrouillet et al., 2007; Kyllonen & Zu, 2016).

### 4.2 RESULTS

**Human Performance**   Human participants achieve accuracies of 72%, 87% and 85% across the 402 examples. Of all the examples, 241 of them are answered correctly by all three participants, 115 are answered correctly by two and 46 questions are answered correctly by one or none. Response times exhibit moderate correlation among participants, with a Pearson correlation coefficient exceeding 0.4. Additionally, the intraclass correlation coefficient ($ICC2 = 0.529$) indicates moderate agreement, and the average response time is deemed reliable with $ICC2K = 0.771$, reflecting good consistency across participants. We also investigate the relationship between response correctness and average

---

[1]We name o1, o3, o4-mini with prefix 'GPT' to indicate model family.

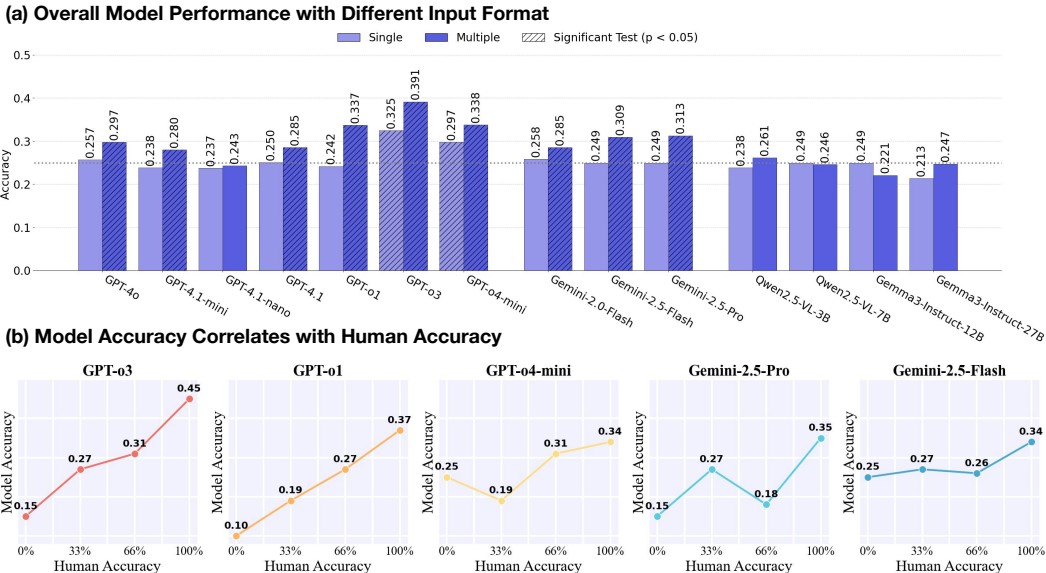

Figure 3: **(a)** Models perform better with multiple separate images as input compared to a single image. With multiple-image input, most closed-source models pass the significance test ($p < 0.05$) over random guess, whereas still all open-sourced models fail. **(b)** MLLM performance correlates with human accuracy (0-3 correct responses across all participants), achieving higher accuracy on instances rated as easier by human evaluators.

response time, showing an inverse correlation ($Pearson = -0.284$). This reveals that questions with higher accuracy tend to elicit shorter response times.

**Overall Model Performance** We present a comprehensive overview of the performance of all evaluated MLLMs in Figure 3(a). This includes a direct comparison of accuracies achieved under both the single composite image and the separate images task settings. The results highlight significant variability in performance, not only between different models but also across the two distinct evaluation paradigms. Closed-sourced models generally achieve higher accuracy than open-source models. Within open-source models, there is no significant performance difference based on model size; all open-sourced models perform comparably to a randomly sampled baseline. Furthermore, we investigate whether model response length correlates with accuracy, analogous to trends observed in human performance. Using Gemini 2.5 Pro which provides token-level counts for both internal reasoning ("thinking") and final response, we measure the Pearson correlation between response length and accuracy. The resulting correlation coefficient is 0.021, indicating no meaningful relationship between the two and suggesting that, unlike in humans, longer responses do not necessarily imply more accurate reasoning in the model.

Detailed per-model and per-task results are provided in Tables 4 and 5. Table 6 further breaks down performance by cognitive capability for proprietary models, showing that most struggle with Flexibility of Closure, which requires identifying hidden patterns in images. Compared to Spatial Relation and Orientation, models show weaker performance on Spatial Visualization tasks in general. Qualitative analyses in Figure 8 corroborate these findings: reasoning traces reveal frequent failures to identify hidden patterns with much visual information loss or anticipate the consequences of spatial manipulations, ultimately leading to incorrect predictions.

**Critique of Single Composite Image Evaluation** Our findings indicate a notable discrepancy in model performance between the two evaluation settings in advanced models. Specifically, the single composite image approach consistently yielded lower accuracies by 4% on average across GPT series models compared to the separate images setting. Most closed-source models significantly

outperformed a random baseline ($p < 0.05$) when using separate images, whereas only GPT o3 and o4-mini showed significant difference from the baseline with a single composite image input. This observation suggests that the challenge in the single image setup may stem more from the complexities of parsing cluttered visual components and segregating distinct conceptual entities, rather than purely from a deficiency in spatial reasoning. Consequently, we posit that previous benchmarks employing solely this composite image methodology do not accurately reflect the intrinsic spatial cognition capabilities of current MLLMs. Therefore, we only discuss evaluation results with separate images as input in the following sections.

**Models are more likely to success on instances that humans perceive as easier.** We investigate whether MLLM performance is essentially random across different complexity levels reflected by human performance. Figure 3(b) plots model accuracy against average human accuracy for the same subset of examples, revealing a general upward trend: models tend to perform better on instances that humans also find easier, indicated by positive slopes. This correlation, supported by statistically significant tests against a random baseline, suggests that current MLLMs do exhibit early signs of spatial cognition. While their reasoning remains limited, the non-random variation in performance across difficulty levels justifies the presence of spatial cognition in these models.

### 4.3 DISCUSSION AND ANALYSIS

Building on our high-level performance analysis, we investigate whether instance-wise correctness can be predicted from relevant features. This is important for assessing the reliability of MLLMs: if consistent patterns exist, model correctness can be anticipated, enabling safer and more robust deployment (Zhou et al., 2023). Comparison with humans further reveals how closely MLLMs mirror human-like spatial reasoning and help to guide model development.

**Qualitative Analysis of Failure Modes** We performed a systematic qualitative error analysis over the responses from Gemini 2.5 Pro, GPT-4o with separate images, and identified four recurring failure modes: **1)** *Perceptual error*: Difficulty representing fine-grained visual primitives (repetition count, small shape details or spatial structures) during reasoning. This often leads to symbolic approximations in ASCII format (e.g. X O O | X X O | ...) or with referable textual descriptions (e.g. Y-pentomino) that omit critical information. **2)** *Manipulation error*: Incorrect or incomplete elicitation of the outcomes about spatial manipulations such as rotations, reflections, or multi-step spatial operations. This common failure mode is more severe when it comes to complex operations such as 3D rotations or symbol tagging, which in the end leads to inconsistent or wrong reasoning chains. **3)** *Spatial relation analysis error*: High-level recognition of certain spatial relations (2D rotation and 2D translation) but failure with other spatial relations. In addition, constrained by perceptual and manipulation errors, we observe inconsistent alignment between the identified spatial relations and precise local features. **4)** *Cascaded logical error*: Early perceptual or manipulation mistakes often lead models to conclude that "no option is correct", then guess. These failure modes align with our cognitive features covering perception and reasoning, and we conduct quantitative analyses to explain model behavior and identify the most impactful features.

**Quantitative Analysis Setups** To explore how well perceptual and reasoning features can explain behavior (cognitive profile), we use machine learning classifiers (random forest) to predict instance-level correctness for both humans and MLLMs. To address label imbalance, class weights are adjusted inversely to class frequencies in the input data when training the classifier. We consider two classification settings: binary classification (correct vs. incorrect) and four-class classification (0-3 correct responses across participants). To further analyze cognitive effort, we treat both as approximations for test-time computational effort: response time for human cognitive load, token count for model reasoning effort. We apply linear regression to predict human response time and MLLM token counts including thinking using the same set of features. The cognitive-related input features are introduced as follows, encompassing both perceptual and reasoning-related dimensions.

For visual perception, we include three features: the pattern complexity of both the question and the answer options, as well as the image resolution. Image resolution can impact perceptual recognition, with lower fidelity obscuring visual structure, so we discretize resolution into three bins (low, medium, high) to reflect practical perceptual clarity. For general reasoning, we extract four features representing the number of reasoning steps required for each category of atomic operations: *Pattern Matching*,

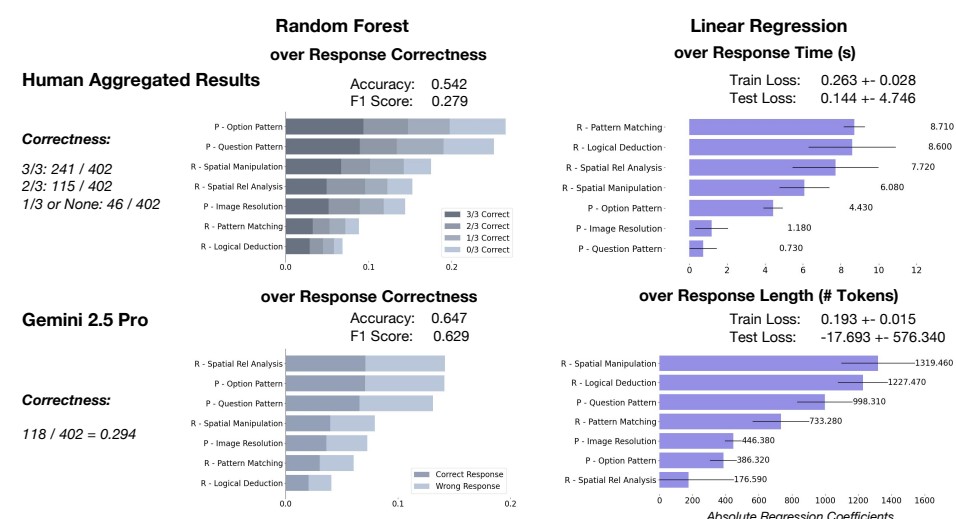

Figure 4: Cognitive profile analysis using SHAP values for correctness prediction and linear regression coefficients for cognitive load, comparing humans and MLLMs. More results in Figures 6 and 7.

*Spatial Relation Analysis*, *Spatial Manipulation*, and *Logical Deduction*. To ensure a stable signal from human, in addition to the correctness of individual human participant, we aggregated responses from three evaluators, as individual responses may be subject to idiosyncratic noise preventing reliable modeling of human cognitive profiles, while models are largely deterministic.

**Human correctness is predictable while MLLMs exhibit near-random instance-level behavior.**
We train the classifiers over the set of examples for human evaluation for fair comparison between human and models using 5-fold cross-validation. Our goal is not to maximize classification accuracy, but to identify the presence or absence of structured cognitive profiles. To mitigate the effects of severe data imbalance and limited samples per fold due to high human accuracy and low model accuracy, we aggregate predictions across folds for more stable metric estimation. Human correctness of individual participants is highly predictable with Random Forest, reaching weighted F1 scores of 0.631, 0.821 and 0.799 ($p < 0.0002$) and AUC score of 0.579, 0.643 and 0.621. In the more granular four-class setting (aggregated human correctness), the classifier still performs above chance (F1 = 0.279 vs. 0.192, $p < 0.05$), reinforcing the presence of systematic cognitive behavior. In contrast, classifiers trained on MLLM outputs fail to detect consistent correctness patterns. As shown in Figure 7, weighted F1 scores and AUC scores remain lower than human participants across most model variants, with no significant improvement over random baselines ($p > 0.01$). These results suggest that human responses are governed by predictable cognitive strategies, while current MLLMs lack the internal structure for reliable spatial reasoning at instance level.

**Pattern complexity drives human correctness, while reasoning features govern cognitive effort.**
To understand which features contribute most to human success, we apply SHAP analysis to the trained classifiers. As shown in Figures 4 and 6, *Pattern Complexity* (especially in answer options) is the strongest predictor of correctness across all participants. This is followed by the presence of *Spatial Manipulation*, a cognitively demanding reasoning step. We further model human response time, a proxy for cognitive effort, using linear regression on the same features. The model predicts time with average error <1 second (±4s), and analysis of coefficients shows that reasoning features (*Spatial Relation Analysis*, *Spatial Manipulation*, *Logical Deduction*) are the dominant contributors to increased response time. Interestingly, *Pattern Matching* correlates with shorter response times, possibly due to heuristic strategies such as visual elimination or rule-of-thumb matching. Together, these results highlight a dual cognitive profile in humans: while perceptual errors (e.g., misreading complex patterns) drive most mistakes, reasoning complexity governs cognitive effort.

**MLLMs show partial alignment with human profiles, but responses remain sensitive to low-level visual cues.** We apply SHAP analysis to the classifiers trained on MLLM correctness (Figure 7) and observe high variability across models, with most failing to reach statistical significance (denoted

in *orange* with $p > 0.01$). Still, some convergence with human cognition emerges. *Option Pattern Complexity* is a shared influential feature across both humans and MLLMs, while features like *Image Resolution* and *Spatial Relation Analysis* are more prominent for certain MLLMs. This suggests that while models do attend to meaningful patterns, they remain disproportionately influenced by low-level visual cues and spatial relationship understanding. To further investigate MLLM effort, we model "thinking length" using linear regression. Here, we find that in addition to reasoning-related features, *Question Pattern Complexity* contributes significantly, while *Spatial Relation Analysis* appears to be the least predictive factor, marking a clear divergence from human profiles. These findings point to a hybrid picture: while MLLMs exhibit emerging spatial awareness, their instance-level reasoning remains noisy and constrained by understanding low-level visual cues, calling for further research.

## 5 CONCLUSION

This work introduced a novel framework with 11PLUS-BENCH benchmark for evaluating MLLMs' spatial cognition against human cognitive profiles, moving beyond aggregate accuracy to fine-grained analysis with predictive power. Our findings show that while current MLLMs show early signs of spatial reasoning, their overall capabilities remain limited with randomness. Human accuracy is consistently shaped by pattern complexity and reasoning demands, revealing structured and predictive cognitive profiles. In contrast, model behavior is more influenced by understanding low-level visual cues such as image resolution and spatial relations, with less predictable and interpretable responses at instance level. These results highlight both emerging capabilities and critical gaps between human and MLLMs spatial cognition. We hope our findings and 11PLUS-BENCH benchmark with finegrained cognitive feature annotations serve as a foundation for future research toward closing this gap, enabling the development of MLLMs with more robust, human-aligned spatial capabilities.

## ETHICS STATEMENT

Our research adheres to rigorous ethical guidelines. We verified the licenses of all softwares and datasets we used in this study and ensured full compliance with their terms. During the human annotation and evaluation process, all annotators provided informed consent for their data to be used as part of the project without any personally identifiable information being collected throughout the process. No privacy concerns have been identified. Furthermore, we have thoroughly assessed the project and do not anticipate any additional potential risks. The study complies with the ICLR Code of Ethics and relevant legal standards, and dataset releases will include documentation to ensure transparency and reproducibility.

## REPRODUCIBILITY STATEMENT

Appendix B details the data collection and processing steps. Experimental setups are provided in Appendix C. All data and inference scripts will be released publicly upon acceptance to facilitate reproducibility.

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

## A  THE USE OF LARGE LANGUAGE MODELS

Large language models (LLMs) were used as general-purpose tools in this work. Specifically, LLMs assisted in polishing the writing to improve clarity and readability.

## B  11PLUS-BENCH

**Overview of the Framework**   We introduce an evaluation framework designed for a fine-grained analysis of MLLMs' spatial reasoning capabilities. The framework extends beyond previous benchmarks in three crucial ways.

*1. Disentangling Cognitive Features (§3.2).* Previous benchmarks often conflate distinct cognitive features that affect model accuracy in spatial reasoning tasks, such as perceptual difficulty and inherent reasoning complexity. Ignoring these features undermines evaluation validity and explainability, hindering real-world applicability when selecting appropriate models (Burden et al., 2023). Our framework explicitly identifies and accounts for these performance-affecting features:

- *Visual Perception*: Complex visual patterns require accurate interpretation of pattern structures before reasoning begins.

- *General Reasoning*: The inherent complexity of the reasoning process itself, e.g., requiring multiple reasoning hops or intricate spatial transformations, adds difficulty that might overshadow an MLLM's genuine spatial reasoning capabilities.

*2. Instance-Wise Evaluation with Predictive Power (§4.2).* Typical average-based benchmark scores (e.g., accuracy) primarily represent overall performance, making it difficult to anticipate whether a model will correctly answer a new question. Inspired by Zhou et al. (2025), our framework enhances interpretability by supporting instance-wise evaluation. This allows researchers to estimate the likelihood that a model will correctly answer a given question based on known cognitive features (Burden et al., 2023), informing both deployment decisions and future research directions.

*3. Parallel Analysis with Human Cognitive Profiles (§4.2).* Despite drawing inspiration from human cognitive tests, previous work lacks direct comparison with human cognition. We bridge this gap by incorporating human evaluation with *response time* for each question as a proxy for human-perceived task difficulty (Barrouillet et al., 2007; Kyllonen & Zu, 2016). This parallel analysis reveals the extent

to which current MLLMs emulate or diverge from human-like spatial cognition, offering insights to guide the advancement of MLLMs.

This dataset is for research purposes only and should not be used outside of research contexts.

**Data Source**   We construct the benchmark from two primary sources: a public subset collected from the web and a private subset sourced from purchased educational materials. For the public data, we crawl the web using carefully selected spatial reasoning keywords. For the private dataset, we acquire spatial aptitude test materials from certified test preparation providers, targeting children under 11 years old.

To ensure the quality of the crawled data and retain only well-formed spatial problems, we implement a filtering pipeline that discard repetitive items based on the urls and ask human annotators to filter out samples that are irrelevant, ambiguous or do not evaluate spatial reasoning. All the data is expressed in English.

**Targeted Capabilities and Task Types**   We focus on spatial cognition tasks designed for young adolescents, using the 11+ exam level as an anchor. Given that not all spatial cognitive skills are equally suited for evaluation in MLLMs, we concentrate on the following three core capabilities: *Spatial Relation and Orientation*, *Spatial Visualization* and *Flexibility of Closure*. Each capability encompasses a collection of tasks, with definitions and examples summarized in Table 1. The selected tasks emphasize interpretable reasoning steps and perceptual challenges amenable to MLLM analysis.

**Expert Annotation Protocol**   We recruit three domain experts to annotate the benchmark data. All annotators hold postgraduate degrees or higher in STEM fields, with backgrounds in mathematics or engineering. The annotation process adheres to institutional ethical guidelines. All annotations are collected anonymously and no information that names or uniquely identifies individual people or offensive content are collected or used. The instructions explain that the data would be used for research purpose only.

**Annotation Fields and Guidelines**   As described in Section 3.2, all samples are annotated for two cognitive dimensions: *Visual Perception Complexity* and *General Reasoning Process*.

For tasks with highly standardized visual transformations, such as 2D shape rotation, 2D shape reflection, or 3D-2D view, we do not require annotators to document full reasoning steps, as these processes are straightforward and consistent across samples. For all other tasks, each expert independently provides both visual perception and reasoning annotations according to the detailed protocol described below.

*Visual Perception Complexity*   We quantify visual complexity for both the question and the option choices. The complexity score is derived from the number of atomic components in each pattern. We define atomic components as key features:

- For referable shapes (e.g., heart, star), complexity is based on the number of symbolic elements.
- For abstract 2D patterns, we count the number of lines or segments.
- For abstract 3D structures, we count the number of surfaces or faces.

This methodology yields a consistent, interpretable complexity score for each visual input. Example annotations are shown in Figure 2.

*General Reasoning Process*   To capture the reasoning process, we define a taxonomy of atomic operations that cover a wide range of spatial reasoning strategies. Annotators must select one operation per step from the categories defined below:

*Pattern Matching*: Determine whether one entity visually contains or resembles another. The match can be based on exact visual similarity or shared key features. Shape matching does not involve reasoning about spatial relationships, nor does it alter the spatial properties of the entities involved.

```
def pattern_match(entity_a: Object, entity_b: Object) -> bool
```

*Spatial Relation Analysis*: Analyze the spatial relationship between two entities. Any two non-overlapping 2D or 3D shapes can be treated as separate entities, for example, two cubes, or two faces

Table 1: Spatial capabilities and corresponding tasks, with question descriptions and number of examples in public and private split.

| Capability | Task | Question Description | Public | Private |
|---|---|---|---|---|
| Spatial Relation and Orientation | 2D shape rotation (SRO.1) | The image shows several 2D shapes, including a designated target shape. Select the option that is the target shape rotated to a different orientation. | 35 | 10 |
| | 2D shape reflection (SRO.2) | The image displays several 2D shapes, with one identified as the target shape. The target shape has been reflected across a mirror line shown in the image. | 33 | - |
| | 3D shape rotation (SRO.3) | This image shows a 3D polycube shape. Choose the option that represents the same shape, viewed from a different rotation. | 6 | 3 |
| Spatial Visualization | Shape completion (SV.1) | The image presents an equation involving a target shape and several shape candidates that can be added to or removed from the base shape. | 9 | 10 |
| | Shape combination (SV.2) | The image illustrates an equation involving a basic shape, where shapes are either added or removed. Only edges labeled with the same letter can be combined. | 68 | 10 |
| | Building blocks (SV.3) | The image displays a target complex 3D shape along with several sets of blocks. Identify the set of blocks that can be combined to form the target shape. | 52 | 10 |
| | Paper folding (SV.4) | The image shows a piece of paper being folded and then punched with holes. Select the option that correctly shows the pattern of holes after the paper is fully unfolded. | 229 | 9 |
| | Cube and nets (SV.5) | The image shows an unfolded shape (net) and several cube candidates. Identify which option can be correctly folded into a cube from the given unfolded shape. | 201 | 9 |
| Flexibility of Closure | Hidden shape (FoC) | The target shape is hidden within one of the answer options. It may be rotated and embedded within the option. Identify the option that contains the hidden target shape. | 76 | 10 |
| Comprehensive (SV+SRO) | Cube and dice (Com.1) | The image shows different views of the same cube, with a unique symbol on each of its six faces. Determine which option correctly matches the missing face. | 17 | 10 |
| | 3D-2D view (Com.2) | This image displays a 3D object. Select the option that correctly represents a 2D view of the object from a specific perspective. | 98 | 10 |

of the same cube, depending on the context of analysis. This process does not change the spatial properties or the overall spatial layout of the entities. Subtypes include:

- Position: Determine the relative position of shape B within entity A.
- Orientation: Determine the direction a part of shape A or entity B is facing (e.g., "Part X of A points toward C").
- Perspective: Infer the viewpoint (e.g., "viewed from behind").
- Rotation: Determine the direction or angle of rotation.
- Folding: Determine the direction in which a 2D net folds into a 3D object.
- Projection: Determine the direction in which a 3D entity is projected onto a 2D plane.

```
def spatial_relation(entity_a: Object, entity_b: Object)
-> statement: str
```

*Spatial Manipulation*: Change the spatial properties or overall spatial layout of entities.

- 2D operations: rotation, translation, reflection, adding/removing shapes
- 3D operations: 3D rotation (around an axis), 3D translation, 3D symmetry
- Dimensional transformations: projection in a certain direction, folding along an edge
- Counting: e.g., counting the number of holes in an origami structure
- Symbol tagging: labeling shapes or parts with markers or symbols

```
def spatial_manipulate(entity: Object, statement: str)
-> Union[Object, str]
```

*Logical Deduction*: Infer rules or verify spatial conditions.

- Logical inference: inferring spatial properties or rules, such as:
  - "A cannot be adjacent to B"
  - "A must be opposite to C"
  - "Cube A can be obtained from Cube B via one or two rotations"
- Verification: testing whether a property or rule holds on another entity

```
def logical_deduction(*statements: str) -> Union[str, bool]
```

Annotators are instructed to decompose their reasoning into step-by-step sequences using these operations, ensuring consistency and reproducibility. This structured representation enables us to map human reasoning steps to potential model behaviors. It is worth noting that we intentionally did not impose a target solution chain or a maximum step count. Annotators decomposed each instance according to their own reasoning, following the principle of minimal sufficient decomposition. Importantly, all entities, relationships, and hypotheses used in the reasoning process had to be either provided in the original question/image or introduced in prior reasoning steps.

## C  EXPERIMENTS

### C.1  HUMAN PARTICIPANTS

We recruit three human participants as evaluators to evaluate human performance and record human behavior (response time in seconds). They are not involved in the annotation process with STEM major background for bachelors major, such as Informatics and AI. All the human evaluators are gathered physically to conduct human evaluations, making sure that the performance really reflects their abilities and behaviors.

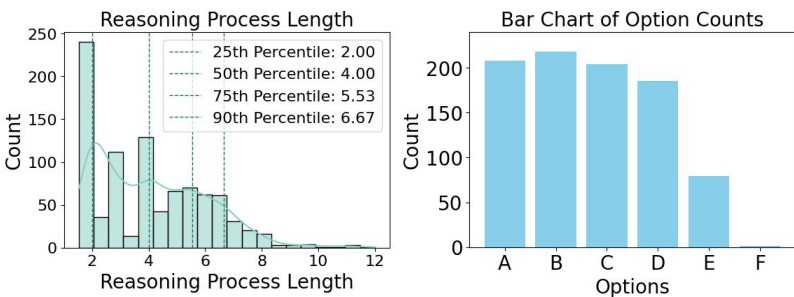

Figure 5: Data distributions over lengths of reasoning process and golden options.

Table 2: Prompt templates for main experiments with single image as input.

---

**Single Image Input**

<QUESTION>
<image>
Conclude your chosen answer to the multiple-choice question between <ANSWER> and
</ANSWER>.

---

## C.2 MODELS

**Hyperparameters**   We adopt most of the inference parameters by default for proprietary models.
For open-sourced models, we adopt the default configuration in HuggingFace. Data and inference
scripts will be released upon acceptance.

**Prompts**   Table 2 and 3 show the prompt templates for single image setting and separate image
setting respectively. Within the prompt templates, <QUESTION> and <OPTIONS> are replaced
with the questions in Table 1 for different tasks.

## C.3 RESULTS

A detailed breakdown of scores per model and per task category is provided in Table 4 and 5 for
multiple separate images and single image as input.

Figure 6 and 7 present extended cognitive pattern analyses across individual human participants and
a broader set of MLLM variants. For human participants, *Pattern Complexity* consistently ranks as
the most influential factor for correctness, while *Logical Deduction* and *Pattern Matching* appear

Table 3: Prompt templates for main experiments with separate image segments as input.

---

**Separate Image Input**

<QUESTION>
<image>
A: <image>
B: <image>
C: <image>
D: <image>
E: <image>
Conclude your chosen answer to the multiple-choice question between <ANSWER> and
</ANSWER>.

---

Table 4: Task-wise performance per model with separate multiple images as input.

| Model | SRO.1 | SRO.2 | SRO.3 | SV.1 | SV.2 | SV.3 | SV.4 | SV.5 | FoC | Com.1 | Com.2 |
|---|---|---|---|---|---|---|---|---|---|---|---|
| GPT 4o | 0.267 | 0.485 | 0.444 | 0.158 | 0.128 | 0.290 | 0.357 | 0.257 | 0.279 | 0.222 | 0.370 |
| GPT 4.1-mini | 0.289 | 0.273 | 0.333 | 0.368 | 0.295 | 0.194 | 0.340 | 0.248 | 0.279 | 0.074 | 0.278 |
| GPT 4.1-nano | 0.200 | 0.394 | 0.444 | 0.211 | 0.192 | 0.387 | 0.269 | 0.195 | 0.163 | 0.185 | 0.269 |
| GPT-o1 | 0.378 | 0.364 | 0.444 | 0.158 | 0.205 | 0.258 | 0.445 | 0.338 | 0.256 | 0.222 | 0.324 |
| GPT-o3 | 0.444 | 0.485 | 0.556 | 0.316 | 0.295 | 0.274 | 0.458 | 0.448 | 0.349 | 0.185 | 0.306 |
| GPT-o4-mini | 0.267 | 0.485 | 0.444 | 0.263 | 0.231 | 0.452 | 0.395 | 0.305 | 0.349 | 0.185 | 0.306 |
| Gemini 2.0 Flash | 0.222 | 0.212 | 0.444 | 0.158 | 0.179 | 0.323 | 0.382 | 0.257 | 0.267 | 0.185 | 0.278 |
| Gemini 2.5 Flash | 0.356 | 0.242 | 0.444 | 0.211 | 0.269 | 0.339 | 0.395 | 0.276 | 0.174 | 0.296 | 0.315 |
| Gemini 2.5 Pro | 0.333 | 0.394 | 0.222 | 0.263 | 0.308 | 0.323 | 0.378 | 0.300 | 0.128 | 0.296 | 0.324 |
| Open-Sourced Models | | | | | | | | | | | |
| Qwen 2.5VL 3B | 0.267 | 0.182 | 0.333 | 0.158 | 0.295 | 0.387 | 0.235 | 0.276 | 0.198 | 0.259 | 0.278 |
| Qwen 2.5VL 7B | 0.133 | 0.424 | 0.111 | 0.211 | 0.218 | 0.387 | 0.218 | 0.214 | 0.209 | 0.407 | 0.306 |
| Gemma3 12B | 0.289 | 0.212 | 0.333 | 0.316 | 0.154 | 0.242 | 0.265 | 0.205 | 0.209 | 0.185 | 0.157 |
| Gemma3 27B | 0.178 | 0.303 | 0.333 | 0.211 | 0.192 | 0.258 | 0.324 | 0.238 | 0.128 | 0.259 | 0.231 |

Table 5: Task-wise performance per model with single images as input.

| Model | SRO.1 | SRO.2 | SRO.3 | SV.1 | SV.2 | SV.3 | SV.4 | SV.5 | FoC | Com.1 | Com.2 |
|---|---|---|---|---|---|---|---|---|---|---|---|
| GPT 4o | 0.156 | 0.364 | 0.333 | 0.368 | 0.256 | 0.371 | 0.248 | 0.224 | 0.244 | 0.407 | 0.231 |
| GPT 4.1-mini | 0.111 | 0.242 | 0.556 | 0.211 | 0.167 | 0.290 | 0.265 | 0.214 | 0.291 | 0.185 | 0.250 |
| GPT 4.1-nano | 0.267 | 0.303 | 0.556 | 0.105 | 0.218 | 0.323 | 0.311 | 0.186 | 0.221 | 0.185 | 0.130 |
| GPT-o1 | 0.200 | 0.364 | 0.444 | 0.211 | 0.167 | 0.242 | 0.261 | 0.238 | 0.233 | 0.185 | 0.250 |
| GPT-o3 | 0.378 | 0.273 | 0.444 | 0.158 | 0.282 | 0.306 | 0.382 | 0.400 | 0.221 | 0.148 | 0.231 |
| GPT-o4-mini | 0.311 | 0.394 | 0.222 | 0.211 | 0.218 | 0.339 | 0.332 | 0.300 | 0.291 | 0.148 | 0.278 |
| Gemini 2.0 Flash | 0.178 | 0.242 | 0.333 | 0.211 | 0.128 | 0.435 | 0.298 | 0.276 | 0.256 | 0.111 | 0.204 |
| Gemini 2.5 Flash | 0.178 | 0.242 | 0.333 | 0.211 | 0.128 | 0.435 | 0.298 | 0.276 | 0.256 | 0.111 | 0.204 |
| Gemini 2.5 Pro | 0.267 | 0.333 | 0.333 | 0.263 | 0.205 | 0.323 | 0.387 | 0.410 | 0.279 | 0.296 | 0.259 |
| Open-Sourced Models | | | | | | | | | | | |
| Qwen 2.5VL 3B | 0.267 | 0.212 | 0.222 | 0.211 | 0.269 | 0.194 | 0.227 | 0.276 | 0.279 | 0.185 | 0.176 |
| Qwen 2.5VL 7B | 0.333 | 0.212 | 0.111 | 0.211 | 0.321 | 0.435 | 0.235 | 0.229 | 0.174 | 0.111 | 0.250 |
| Gemma3 12B | 0.156 | 0.212 | 0.222 | 0.316 | 0.282 | 0.371 | 0.231 | 0.229 | 0.256 | 0.370 | 0.241 |
| Gemma3 27B | 0.200 | 0.212 | 0.111 | 0.211 | 0.244 | 0.274 | 0.227 | 0.224 | 0.221 | 0.111 | 0.139 |

Table 6: Capability-wise performance for proprietary models with separate multiple images as input.

| | SRO | SV | FoC | Comprehensive |
|---|---|---|---|---|
| GPT 4o | 0.368 | 0.280 | 0.279 | 0.341 |
| GPT 4.1-mini | 0.287 | 0.288 | 0.279 | 0.237 |
| GPT 4.1-nano | 0.299 | 0.244 | 0.163 | 0.252 |
| GPT-o1 | 0.379 | 0.349 | 0.256 | 0.304 |
| GPT-o3 | 0.471 | 0.410 | 0.349 | 0.282 |
| GPT-o4-mini | 0.368 | 0.344 | 0.349 | 0.282 |
| Gemini 2.0 Flash | 0.241 | 0.300 | 0.267 | 0.259 |
| Gemini 2.5 Flash | 0.322 | 0.326 | 0.174 | 0.311 |
| Gemini 2.5 Pro | 0.345 | 0.333 | 0.128 | 0.319 |

Table 7: Response format parsing result with single image as input.

| Model | Success | Ordinal | Number | Letter | Unknown | Verbalized Choice | Parsing Failure |
|---|---|---|---|---|---|---|---|
| GPT 4o | 843 | 10 | 34 | - | 27 | - | 1 |
| GPT 4.1-mini | 846 | 14 | 43 | 3 | 9 | - | - |
| GPT 4.1-nano | 801 | 3 | 50 | 16 | 25 | 20 | |
| GPT-o1 | 852 | - | 31 | 3 | 27 | 1 | 1 |
| GPT-o3 | 862 | 1 | 34 | 1 | 16 | - | 1 |
| GPT-o4-mini | 818 | - | 33 | 5 | 21 | 34 | 3 |
| GPT 4.1 | 855 | 6 | 30 | - | 24 | - | - |
| Gemini 2.0 Flash | 728 | 12 | 19 | 4 | 23 | 129 | - |
| Gemini 2.5 Flash | | | | | | | |
| Gemini 2.5 Pro | | | | | | | |
| Qwen 2.5VL 3B | 814 | - | 22 | 15 | 48 | 13 | 3 |
| Qwen 2.5VL 7B | 829 | - | 26 | 18 | 39 | 3 | - |
| Gemma3 12B | 824 | 1 | 55 | 1 | 11 | 22 | 1 |
| Gemma3 27B | 817 | 5 | 44 | 10 | 37 | 1 | 1 |

Table 8: Response format parsing result with separate multiple images as inputs.

| Model | Success | Ordinal | Number | Letter | Unknown | Verbalized Choice | Parsing Failure |
|---|---|---|---|---|---|---|---|
| GPT 4o | 901 | - | 7 | - | 2 | 1 | 4 |
| GPT 4.1-mini | 900 | - | 12 | 2 | - | - | 1 |
| GPT 4.1-nano | 866 | - | 18 | 13 | 6 | 12 | - |
| GPT-o1 | 903 | - | 7 | 2 | 2 | - | 1 |
| GPT-o3 | 910 | - | 2 | 1 | - | 1 | 1 |
| GPT-o4-mini | 863 | - | 11 | 1 | - | 38 | 2 |
| GPT 4.1 | 898 | - | 13 | 1 | 3 | - | - |
| Gemini 2.0 Flash | 642 | - | - | - | - | 273 | - |
| Gemini 2.5 Flash | - | - | - | - | - | 909 | 5 |
| Gemini 2.5 Pro | | | | | | | |
| Qwen 2.5VL 3B | 752 | - | 5 | 11 | 28 | 119 | - |
| Qwen 2.5VL 7B | 848 | - | 5 | 39 | 22 | 1 | - |
| Gemma3 12B | 881 | - | 1 | 3 | - | 28 | 2 |
| Gemma3 27B | 903 | - | 6 | - | 2 | 4 | - |

less impactful. Moreover, reasoning-related features contribute most significantly to response time, whereas perceptual features such as *Pattern Complexity* and *Image Resolution* are among the least influential in determining response time per sample.

In contrast, classifiers trained to predict MLLM correctness do not significantly outperform a random baseline, as indicated by the orange highlights in Figure 7. No consistent cognitive profiles emerge across model variants: different features dominate in different models, suggesting a lack of stable, interpretable reasoning strategies in current MLLMs.

**Prompting Formulations**   We conducted additional experiments to examine how prompt formulations influence model accuracy using three alternative prompt settings:

- Verbal variations (Verbal Var): Reformulating the question text ("Which option image matches the correct pattern?")
- CoT prompting ("Think step by step and describe the spatial transformations before answering.")
- Hint: Prompts augmented with annotated human reasoning operation types (Appendix B).

The experiments are all conducted with separate multiple images as inputs. As shown in Table 9 and Table 10, across GPT-4o and o3 as representative non-reasoning and reasoning models, we

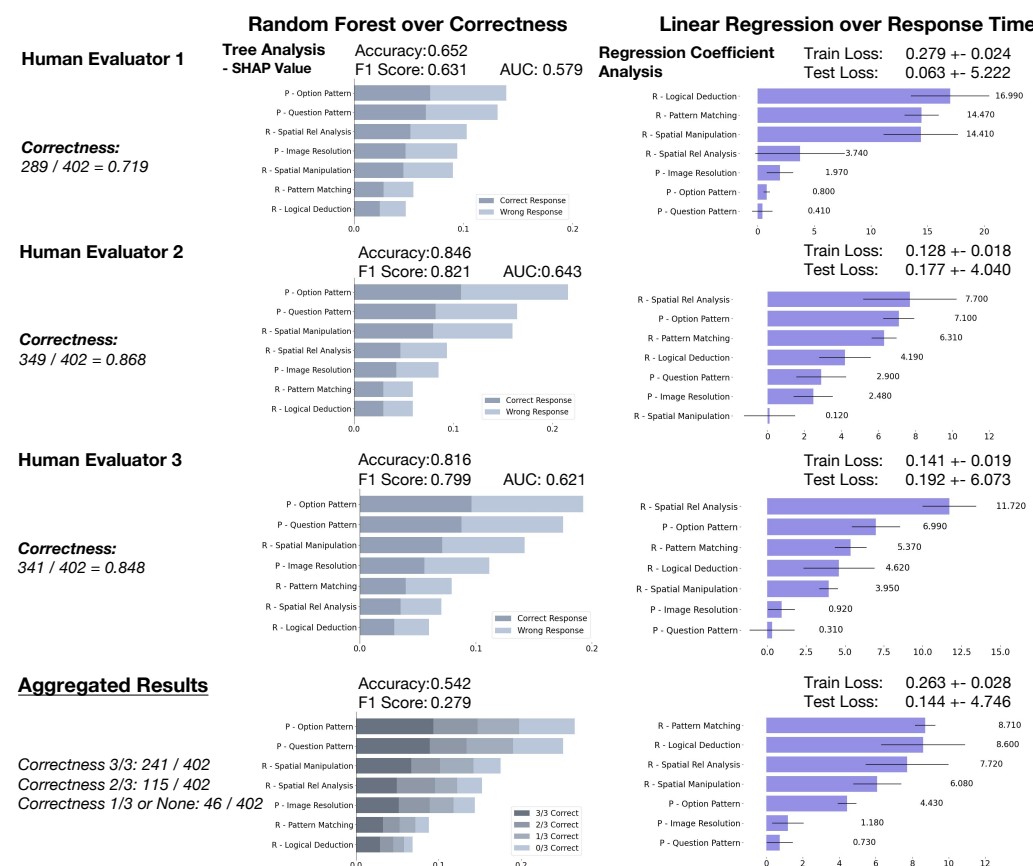

Figure 6: Feature Relevance in the Cognitive Profiles of Individual Human Participants and Aggregated Human Behavior. Individual human responses are predictable with $p < 0.0002$ for F1 score compared to random chance.

Table 9: Model Performance with Different Prompting Formulations

| Model | Original | Verbal Var | CoT | Hint |
|---|---|---|---|---|
| GPT 4o | 0.2973 | 0.3060 | 0.3049 | 0.3027 |
| o3 | 0.3913 | 0.3868 | 0.3945 | 0.3825 |

observe neglectable differences in accuracy relative to our original prompt. Results of o3 align with the findings in OpenAI technical guide (OpenAI, 2025) that reasoning models perform best with straightforward prompts and CoT prompting is unnecessary.

Even provided with hints regarding spatial operations (Hint), the performance is not significantly different, which indicates that the core limitation is the integration of perception, manipulation, and reasoning, rather than a single isolated weakness, aligning with the error modes we identified. Further cognitive-profile analyses remain consistent, with low-level visual cues (specifically pattern complexity and image resolution) followed by spatial relation analysis and spatial manipulation being the most impactful features of model success.

These results indicate that prompt formulation does not significantly affect performance, supporting the robustness of our findings.

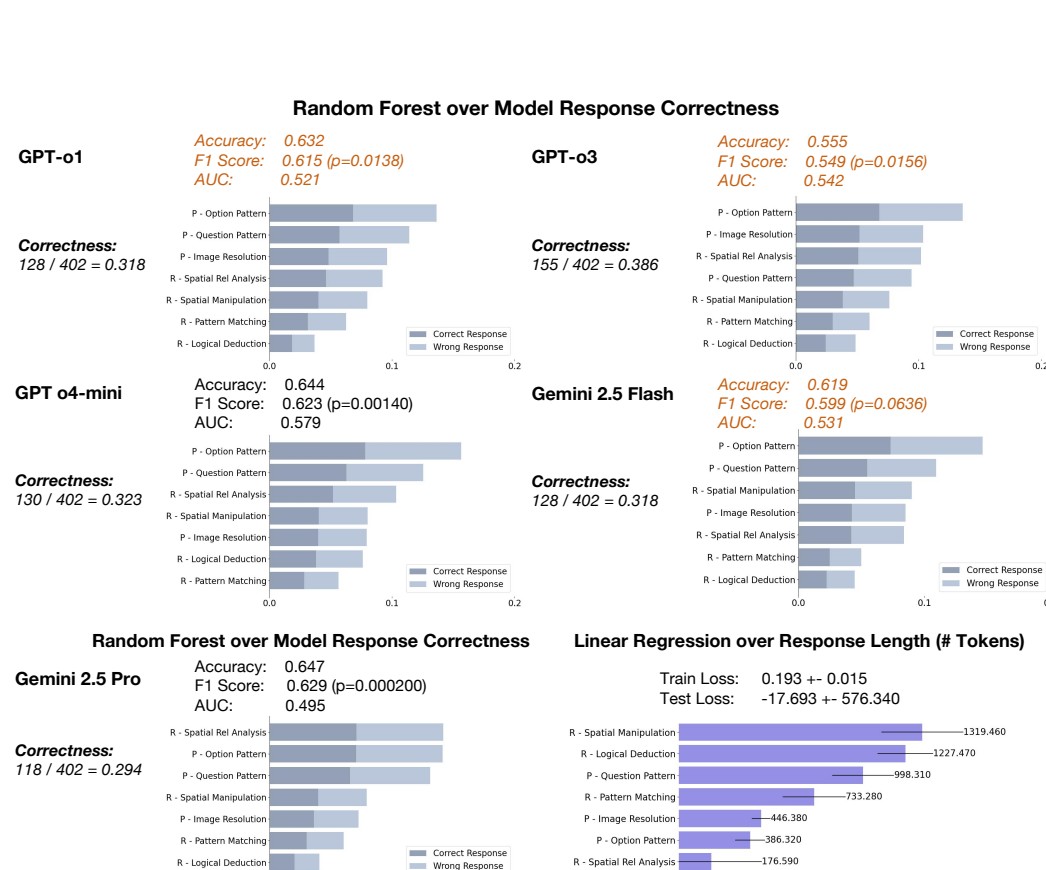

Figure 7: Feature Relevance in the Cognitive Profiles of Different Model Variants.

Table 10: Predictability of Correctness with Different Prompting (* *with p < 0.01*)

| Model | Original | | Verbal Var | | CoT | | Hint | |
|---|---|---|---|---|---|---|---|---|
| | GPT 4o | o3 | GPT 4o | o3 | GPT 4o | o3 | GPT 4o | o3 |
| F1 | 0.622* | 0.549 | 0.595 | 0.585 | 0.658* | 0.580 | 0.620* | 0.564 |
| AUC | 0.561 | 0.542 | 0.462 | 0.575 | 0.583 | 0.579 | 0.569 | 0.598 |
| Acc | 0.647 | 0.555 | 0.622 | 0.600 | 0.677 | 0.587 | 0.647 | 0.572 |

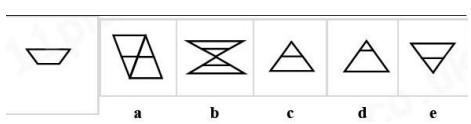

*Textual descriptions only capturing semantic information (isosceles trapezoid with longer base at the top), losing visual information (size, length, etc.)*

The target shape is an isosceles trapezoid with the longer base at the top. Let's examine each option:
A: The middle part of the parallelogram is an isosceles trapezoid with the longer base at the top. This matches the target shape.
...
So, options A, B, and E contain the target shape without rotation. Options C and D contain the target shape if rotated 180 degrees.
......
Let's assume A is the intended answer.

The final answer is $\boxed{A}$.

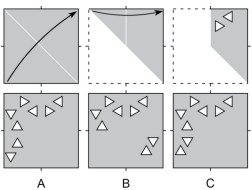

*Struggle to anticipate the consequences of spatial manipulations and refer to symbolic logical deduction in this case, leading to correct number of holes but overlooking the orientation of shapes.*

Let's assume the standard interpretation of the folds, leading to a final folded shape that is 1/4 of the original area and 4 layers thick.
First fold: Fold along the diagonal from bottom-left to top-right. The top-left triangle is folded onto the bottom-right triangle. The resulting shape is the bottom-right triangle of the original square, which is two layers thick. Let the original square be represented by vertices (0,0), (1,0), (1,1), (0,1). The fold line is y=x. The region x<y is folded onto x>y. So the triangle with vertices (0,0)-(0,1)-(1,1) is folded onto the triangle (0,0)-(1,0)-(1,1). The resulting shape is the triangle with vertices (0,0), (1,0), (1,1). This is the bottom-right triangle.
......
Punching: Two triangular holes are punched in the top-left part of this folded triangle. The "top-left" part is near the vertex (0.5,0.5).

Unfolding:
1. Unfold the second fold (along y=-x+1). Each punched hole at (x,y) will be mirrored to (1-y,1-x). Also, the orientation will be reflected.
2. Unfold the first fold (along y=x). Each hole at (x,y) from the previous step will be mirrored to (y,x). Orientation will be reflected.

So, if a hole is at (x,y), after full unfolding, there will be holes at:
Original: (x,y)
Reflected across y=-x+1: (1-y,1-x)
Reflected across y=x: (y,x)
Reflected across y=x then y=-x+1: (1-x,1-y) (reflection of (y,x) across y=-x+1) So, each punched hole (x,y) generates a set of 4 holes: (x,y), (y,x), (1-y,1-x), (1-x,1-y).
Since there are two punched holes, there will be 8 holes in total, unless there are overlaps or symmetries. ....... The final answer is $\boxed{A}$.

Figure 8: Qualitative analysis over the model response (from Gemini 2.5 Pro).

