# OpenReview forum: "11Plus-Bench: Demystifying Multimodal LLM Spatial Reasoning with Cognitive-Inspired Analysis"
_ICLR.cc/2026/Conference — Submitted to ICLR 2026_

### Official Review · Reviewer_xG19 · 2025-10-24

**Soundness:** 3
**Presentation:** 3
**Contribution:** 2
**Rating:** 4
**Confidence:** 4

**Summary:**

The paper introduces 11PLUS-BENCH, a new benchmark for evaluating spatial reasoning in Multimodal Large Language Models (MLLMs), grounded in standardized human spatial aptitude tests (e.g., 11+ exams). Unlike prior benchmarks that report only aggregate accuracy, this work provides fine-grained cognitive annotations—including perceptual complexity (e.g., pattern component counts), reasoning steps (e.g., spatial manipulation, logical deduction), and image-level bounding boxes. The authors conduct human evaluations (N=3) with response time as a proxy for cognitive load and compare performance across 14 MLLMs (open- and closed-source).

Key contributions include:

1. A cognitively inspired evaluation framework enabling instance-level, parallel analysis of human and MLLM "cognitive profiles."
2. Empirical findings showing that while MLLMs exhibit non-random performance correlated with human difficulty, their instance-level correctness is unpredictable and driven more by low-level visual features (e.g., image resolution) than abstract reasoning.
3. Evidence of divergent cognitive mechanisms: human accuracy is shaped by pattern complexity, whereas MLLM behavior is noisy and sensitive to superficial cues.
4. A critique of the common “single composite image” evaluation format, showing that separate image inputs yield more reliable assessment of spatial reasoning.

The work concludes that current MLLMs show early but brittle signs of spatial cognition, lacking the structured, robust reasoning seen in humans.

**Strengths:**

The paper is highly original in its integration of cognitive science methodology into MLLM evaluation. Rather than treating spatial reasoning as a monolithic skill, it decomposes it using psychometrically validated constructs (e.g., Spatial Visualization, Flexibility of Closure) and introduces expert-annotated cognitive features that disentangle perception from reasoning. The idea of building predictive models of correctness using these features—and comparing their explanatory power across humans and models—is novel and insightful.

The benchmark construction is rigorous: dual sourcing (public + proprietary), expert annotation by STEM-trained individuals, high inter-annotator agreement, and contamination controls (e.g., >50% of data lacks public answers). Human evaluation follows cognitive science norms (response time as a cognitive load proxy). The experimental design compares two input formats and includes both open- and closed-source models, enhancing validity. Statistical analyses (SHAP, regression, significance testing) are appropriate and well-executed.

The paper is exceptionally well-structured and clearly written. Figures 1, 4, and 6–8 effectively communicate the framework, results, and qualitative insights. Technical details (e.g., atomic reasoning operations, pattern complexity metrics) are precisely defined in the appendix. The narrative consistently links empirical findings back to the core question: Do MLLMs reason spatially like humans?

This work addresses a critical gap in MLLM evaluation: the lack of human-aligned, process-oriented spatial reasoning benchmarks. By revealing that MLLM performance is unpredictable at the instance level and driven by low-level features, it challenges assumptions of “emergent spatial intelligence” in current models. The benchmark and methodology offer a foundation for future work aiming to build more robust, human-like spatial reasoning systems—relevant to robotics, education, and scientific discovery.

**Weaknesses:**

1. Ambiguity in “Reasoning Steps” Annotation: The paper defines atomic operations (e.g., Spatial Manipulation, Logical Deduction), but it’s unclear how annotators decomposed multi-step problems into sequences. Were reasoning chains unique per instance? How was subjectivity controlled beyond category selection?

2. Underexplored Model “Thinking” Data: The paper notes that Gemini 2.5 Pro provides token-level “thinking” traces, but only uses total token count as a cognitive load proxy. A deeper analysis of content in these traces (e.g., do models verbalize spatial manipulations correctly?) could strengthen claims about reasoning mechanisms.

3. Task Coverage Bias: The benchmark heavily emphasizes 2D paper-folding (229 public examples) and cube nets (201), while other tasks (e.g., 3D rotation) have very few samples (<10). This may skew conclusions about MLLM weaknesses (e.g., in Flexibility of Closure).

**Questions:**

1. Annotation Protocol for Reasoning Chains: For tasks requiring multiple reasoning steps (e.g., paper folding), how did annotators determine the sequence and number of atomic operations? Was there a maximum step limit? Could you share an example of a full annotated reasoning chain vs. what a model generated?

2. Model Architecture Confounds: The paper groups models by open/closed source, but differences in vision encoders (e.g., CLIP vs. proprietary) or training data may explain performance gaps more than “spatial reasoning ability.” Did the authors control for these factors, or consider ablations (e.g., same LLM + different vision backbones)?

3. Contamination Risk in Public Data: While >50% of data lacks golden answers, the public set was crawled from the web using spatial reasoning keywords. Could high-performing closed-source models have seen similar problems during training? How was this risk quantified beyond “no golden answers”?

4. Actionable Pathways: The conclusion states MLLMs lack “compositional understanding.” What specific architectural or training changes does the authors’ analysis suggest? For instance, would explicit spatial operation modules (e.g., differentiable renderers) help, or is the issue more fundamental (e.g., lack of mental simulation)?

---

> ### Author Response · Authors · 2025-11-20
> **Response to Reviewer xG19 (1/2)**
>
> Thank you for your recognition of our work and your valuable suggestions. We would like to address your comments as follows to get more support:
>
> > **W1 & Q1: Annotation Protocol for Reasoning Chains**
>
> Our goal was to capture expert cognitive processes without prescribing a single canonical reasoning chain. To achieve this, we designed an annotation procedure that is **standardized at the atomic-operation level** while allowing **flexibility in decomposition**:
>
> 1) Annotators were trained only on predefined atomic operations (Appendix B), each with a clear input–output protocol, to ensure consistency in operation semantics.
>
> 2) We intentionally did **not** impose a target solution chain or a maximum step count. Annotators decomposed each instance according to their own reasoning, following the principle of minimal sufficient decomposition.
>
> ***Example (Shape Composition)***
>
> For the instance described, a full reasoning chain from an annotator might be:
>
> ```
> Step 1 (Spatial Relation Analysis): SpatialRelationAnalysis(Left-Pattern, Triangle) --> "Triangle should be rotated 90° CW and moved to Left-Pattern."
>
> Step 2 (Manipulation): Manipulation - Rotation (Triangle, 90° CW) --> [Visual Entity: Rotated Triangle]
>
> Step 3 (Manipulation): Manipulation - Translation (Left-Pattern, Rotated Triangle) --> [Visual Entity: Merged Shape]
>
> Step 4 (Pattern Matching): Compare(Merged Shape, Options) --> True.
> ```
>
> It's worth noticing that all entity/relationship/hypothesis used in the reasoning process must be either provided in the original question/image, or proposed in prior reasoning steps.
>
> ***Inter-annotator alignment:***
>
> Our quantitative analysis show both **high-level agreement** with Pearson Correlation of Reasoning Step Length around 0.8 and **subjective variation** as follows, confirming that our annotations are reliable but not rigidly constrained:
>
> * Mean and Variance of average reasoning-step length: 4.66 +- 1.15
>
> * Step-length exact match across annotators: 39.78%
>
> * Mean and Variance of fine-grained operation types: 4.54 +- 0.47
>
> * Fine-grained operation-type exact match: 50.48%
>
> * Strict reasoning-process exact match: 38.41%
>
> These statistics indicate that experts converge on the overall cognitive structure while retaining individual interpretations with a certain level of inherent subjectivity.
>
> > **W2: Underexplored Model “Thinking” Data**
>
> We conducted additional qualitative findings over Gemini 2.5 Pro’s “thinking” traces, indicating that model often verbalize correct spatial operations but fail to align these operations with corresponding visual primitives and tend to hallucinate the executed simulation results. In addition to Gemini 2.5 Pro thinking process, we also performed a systematic qualitative error analysis over reasoning process from GPT-4o with separate images and identified common failure modes as follows:
>
> 1) **Perceptual error**: Difficulty representing fine-grained visual primitives (repetition count, small shape details or spatial structures) during reasoning. This often leads to symbolic approximations in ASCII format (e.g. X O O | X X O | ...) or with referable textual descriptions (e.g. Y-pentomino) that omit critical information.
>
> 2) **Manipulation error**: Incorrect or incomplete elicitation of the outcomes about spatial manipulations such as rotations, reflections, or multi-step spatial operations. This common failure mode is more severe when it comes to complex operations such as 3D rotations or symbol tagging, which in the end leads to inconsistent or wrong reasoning chains.
>
> 3) **Spatial relation analysis error**: High-level recognition of certain spatial relations (2D rotation and 2D translation) but failure with other spatial relations. In addition, constrained by perceptual and manipulation errors, we observe inconsistent alignment between the identified spatial relations and precise local features.
>
> 4) **Cascaded logical error**: Early perceptual or manipulation mistakes often lead models to conclude that “no option is correct” then guess.
>
> These failure modes align with our cognitive-profile findings that low-level visual perception and manipulation are primary impactful features of model performance. We will include the qualitative analysis and representative traces in the paper.
>
> > **W3: Task Coverage Bias**
>
> We acknowledge the distribution bias across different tasks. However, our primary focus is to investigate whether models possess a **consistent cognitive profile with predictive power**, rather than per-task accuracy. Even with uneven task counts, ability categories (e.g., Spatial Visualization, Flexibility of Closure) include sufficient instances to evaluate ability-wise performance and altogether can be used to analyse cross-task cognitive profiles. We believe the present benchmark remains sufficient for modelling generalizable cognitive patterns rather than task-specific performance.

---

> ### Author Response · Authors · 2025-11-20
> **Response to Reviewer xG19 (2/2)**
>
> > **Q2: Model Architecture Confounds**
>
> We agree that fully controlled comparisons (e.g., same LLM with different vision backbones or spatial encodings) would provide cleaner causal attribution. However, such **controlled MLLM families are not publicly available**: current models differ simultaneously in their vision encoders, positional embeddings, base LLMs, and training data. As a result, strict ablations are not feasible with existing systems. In addition, given the **neglectable performance differences** of current open-source models compared with random guess baseline, the performances may not be significantly different with architectural differences as well.
>
> Our grouping by open/closed source was for interpretability, rather than implying openness as the causal factor. Importantly, our cognitive-profile and qualitative analyses focus on consistent modes across diverse models, such as perception and manipulation errors. These cross-model behavioural regularities suggest that the limitations we identify reflect general weaknesses of current models, calling for novel recipes in the future. We will clarify this in the revision and highlight controlled ablations as valuable future work.
>
> > **Q3: Contamination Risk in Public Data**
>
> While a formal contamination audit is challenging without full model pretraining transparency, we have taken multiple precautionary and empirical steps:
>
> 1) **Intrinsic protection** by data collection: Human verification during crawling showed that fewer than 45% of instances contained extractable answers, and these could **not** be reliably obtained through rule-based HTML scraping. The majority of answers therefore required fresh human annotation, creating an inherent barrier against simple automated scraping and data contamination.
>
> 2) Empirical signal: The **low** model performance across all tested MLLMs, despite their web-scale pretraining. We also observe comparable performance on public vs. private sets across models, which argues against contamination-driven advantages. If contamination were substantial, closed-source models would show anomalously high performance on public items (not observed).
>
> 3) Mitigation and evaluation plan in the future: To ensure long-term robustness and considering copyright issues, the private test set will remain non-public and accessible only through a leaderboard-based evaluation. Neither the images nor the answers will be released, preventing future leakage as models evolve.
>
> We believe these combined design and procedural guardrails can help to mitigate contamination risk in the future and ensure reliable generalization results.
>
> > **Q4: Actionable Pathways**
>
> Our cognitive-profile analysis indicates that current MLLMs struggle in accurate grounded visual perception, robust spatial manipulation, and compositional spatial-relation analysis. These findings suggest several potential actionable pathways at both the reasoning and training levels for further research.
>
> (1) Improving grounded visual perception. Our results point to limitations in how models perceive spatial layout and object geometry. Recent progress suggests that strengthening positional and geometric representations can meaningfully improve perception. For example, Qwen2.5-VL incorporates M-RoPE for enriched positional embeddings [1], and G-LLaVA introduces geometric visual–language alignment to enhance geometric awareness [2]. These developments support the view that more grounded, spatially informed representations are a promising direction for achieving grounded perception.
>
> (2) Strengthening spatial manipulation and mental simulation. Our error analysis shows that many failures arise not from misunderstanding instructions, but from the model’s inability to simulate spatial transformations reliably. Early evidence from tool-based spatial reasoning [3] and native multimodal generation approaches (e.g., MVoT [4]) supports this direction: incorporating grounded mental simulation as intermediate steps in visual cues reduces manipulation errors and improves factual consistency. This can be achieved either via external agentic tool-use methods or through architectural advances toward more unified multimodal models, which is left as an open direction for future research.
>
> We hope our benchmark and analysis provide deeper insight into why models fail on specific spatial-reasoning questions and help guide targeted improvements for further research. We'll include this discussion in the revision.
>
> **References**
>
> [1] Bai, Shuai, et al. "Qwen2. 5-vl technical report." arXiv preprint arXiv:2502.13923 (2025).
>
> [2] Gao, Jiahui, et al. "G-llava: Solving geometric problem with multi-modal large language model." ICLR, 2025.
>
> [3] Wu, Junfei, et al. "Reinforcing spatial reasoning in vision-language models with interwoven thinking and visual drawing." NeurIPS, 2025.
>
> [4] Li, Chengzu, et al. "Imagine While Reasoning in Space: Multimodal Visualization-of-Thought", ICML, 2025.

---

### Official Review · Reviewer_AuWm · 2025-10-26

**Soundness:** 2
**Presentation:** 3
**Contribution:** 2
**Rating:** 4
**Confidence:** 4

**Summary:**

11Plus-Bench introduces a cognitively-inspired benchmark for evaluating spatial reasoning in Multimodal Large Language Models (MLLMs), grounded in standardized human spatial aptitude tests. The benchmark features fine-grained expert annotations of perceptual complexity and reasoning steps, enabling parallel, instance-level analysis of human and model cognitive profiles. Experiments across 14 MLLMs and human participants reveal that while models show early signs of spatial cognition, their performance is largely random and sensitive to low-level perceptual features, in contrast to humans whose accuracy is shaped by abstract pattern complexity and structured reasoning.

**Strengths:**

- Fine-Grained Annotation: Provides detailed, expert-annotated cognitive features (perceptual complexity, reasoning steps), enabling nuanced, instance-level analysis.
- Human-Machine Comparison: Enables direct, parallel analysis of human and model cognitive profiles, moving beyond aggregate accuracy to predictive modeling of correctness and cognitive effort - presenting a sound and elaborate testing framework

**Weaknesses:**

- Limited Scale and Diversity: The public set is relatively small (824 examples), and the private set (91) is not open, potentially limiting generalizability and reproducibility.
- Lack of data contamination study :
   - Given the static nature of the dataset and no guardrails/process in place for mitigating/studying data contamination, the benchmark runs the risk of dilution on all metrics provided. There is a need for an elaborate data contamination study and a plan for its mitigation.
    - While the benchmark claims to minimize contamination (using private, non-public test items), there is no empirical audit or adversarial check to quantify or guarantee the absence of contamination in model pretraining data.
- Human Baseline Scope: Human evaluation is limited to three participants, raising questions on reliability
- Qualitative Analysis: The paper could benefit from more qualitative error analysis or case studies to illustrate specific model failure modes.
- Lack of an exhaustive list of models for evaluation - key opensource models like DeepSeek,Kimi and Llama families are missing from evaluation

**Questions:**

Major flags covered in the weakness section

---

> ### Author Response · Authors · 2025-11-20
> **Response to Reviewer AuWm (1/2)**
>
> Thank you for your recognition of our work and your valuable suggestions. We would like to address your comments as follows to get more support:
>
> > **W1: Limited Scale and Diversity**
>
> We agree that a larger and more diverse dataset would further enhance generalizability. Our current focus prioritizes **quality, validity and annotation richness over quantity**.
>
> 1) **Rigorous filtering**: We crawled the raw data with 29 spatial reasoning keywords from the web. We manually filtered the large raw pool (from 5352 webpages and 29 hours of videos) down to 824 items that meet **strict criteria**: no ambiguity, well-defined spatial cognition problems, image quality, as well as potential copyright and privacy issues.
>
> 2) **Fine-grained annotations**: Each item is annotated with multi-level fine-grained cognitive features by three human experts (e.g., bounding boxes, pattern complexity, reasoning process), giving the dataset analytical depth uncommon in MLLM benchmarks.
>
> 3) **Private set** design: For private set fine-grained annotation, we sampled 100 data over 3,000+ candidates and retained only 91 after rigorous quality screening to ensure conceptual validity and high quality. These annotated items and the rest of the private set don't have to be open and can be reserved for controlled leaderboard evaluation if we need it to play the role of detecting leakage and future contamination. We take advantage that this set has **copyright limitations** to keep a private set that can be less prone to contamination in the past, and especially in the future.
>
> Moreover, our thousand-level scale is comparable to widely used reasoning benchmarks (e.g., GPQA [1]: 448 items, GSM8K [2] validation set: 1319 items). In addition, our statistical analyses yield significant human–model correlations and robust predictive results for human participants, supporting the reliability of such benchmark.
>
> > **W2: Lack of Data Contamination Study**
>
> We acknowledge the importance of ensuring data integrity. While a formal contamination audit is challenging without full model pretraining transparency, we have taken multiple precautionary and empirical steps:
>
> 1) **Intrinsic protection** by data collection: Human verification during crawling showed that fewer than 45% of instances contained extractable answers, and these could **not** be reliably obtained through rule-based HTML scraping. The majority of answers therefore required fresh human annotation, creating an inherent barrier against simple automated scraping and data contamination. In addition, the private set has no concern for data contamination since its recency and intellectual property considerations.
>
> 2) Empirical signal: The **low** model performance across all tested MLLMs, despite their web-scale pretraining. We also observe consistent performance on public vs. private sets across models, which argues against contamination-driven advantages. If contamination were substantial, models would show higher performance on public items (not observed).
>
> 3) Mitigation and evaluation plan in the future: To ensure long-term robustness and considering copyright issues, the private test set will remain non-public and accessible only through a leaderboard-based evaluation. Neither the images nor the answers will be released, preventing future leakage as models evolve.
>
> We believe these combined design and procedural guardrails can help to mitigate contamination risk in the future and ensure reliable generalization results.
>
> > **W3: Human baseline scope**
>
> We agree that a larger and more demographically diverse participant pool would further strengthen the study. Our present design involves three expert annotators (for gold answers and cognitive features) and three participants for human evaluation. We relied on expert labour for high-quality fine-grained spatial-cognitive annotations by prioritising quality annotations over quantity. Although constrained by financial budget for human participants, our conclusions remain **robust**:
>
> 1) High cross-participant consistency: We observe strong correlations in response times across participants (Sec. 4.2 - Human Performance).
>
> 2) Stable predictive modelling: Cognitive features reliably predict human correctness across participants (Sec. 4.3).
>
> These results demonstrate **stable behavioural patterns** despite the small sample size of participants, suggesting that our conclusions are not driven by participant-specific variance but by **consistent cognitive signals**.

---

> ### Author Response · Authors · 2025-11-20
> **Response to Reviewer AuWm (2/2)**
>
> > **W4: Qualitative Analysis**
>
> We thank the reviewer for the valuable suggestions. We performed a systematic qualitative error analysis (Gemini 2.5 Pro, GPT-4o with separate images) and identified four recurring failure modes:
>
> 1) **Perceptual error**: Difficulty representing fine-grained visual primitives (repetition count, small shape details or spatial structures) during reasoning. This often leads to symbolic approximations in ASCII format (e.g. X O O | X X O | ...) or with referable textual descriptions (e.g. Y-pentomino) that omit critical information.
>
> 2) **Manipulation error**: Incorrect or incomplete elicitation of the outcomes about spatial manipulations such as rotations, reflections, or multi-step spatial operations. This common failure mode is more severe when it comes to complex operations such as 3D rotations or symbol tagging, which in the end leads to inconsistent or wrong reasoning chains.
>
> 3) **Spatial relation analysis error**: High-level recognition of certain spatial relations (2D rotation and 2D translation) but failure with other spatial relations. In addition, constrained by perceptual and manipulation errors, we observe inconsistent alignment between the identified spatial relations and precise local features.
>
> 4) **Cascaded logical error**: Early perceptual or manipulation mistakes often lead models to conclude that “no option is correct” then guess.
>
> These failure modes align with our cognitive-profile findings that low-level visual perception and manipulation are primary impactful features of model performance. We will include the qualitative analysis to the paper.
>
> > **W5: Lack of an exhaustive list of models for evaluation**
>
> We appreciate the reviewer's suggestion. Our current model selection includes a representative set of both open-source and proprietary models, covering both reasoning and non-reasoning variants, that already exhibit clear **discriminative power** on the performances of spatial reasoning tasks. We believe the models we select are sufficient to reveal consistent behavioural patterns and to support the core findings of our analysis.
>
> **References**
>
> [1] Rein, David, et al. "Gpqa: A graduate-level google-proof q&a benchmark." First Conference on Language Modeling. 2024.
>
> [2] Cobbe, Karl, et al. "Training verifiers to solve math word problems." arXiv preprint arXiv:2110.14168 (2021).

---

### Official Review · Reviewer_yP9B · 2025-10-31

**Soundness:** 3
**Presentation:** 3
**Contribution:** 3
**Rating:** 6
**Confidence:** 3

**Summary:**

This paper introduces 11PLUS-BENCH, a new benchmark inspired by human cognitive tests to evaluate the spatial reasoning of Multimodal Large Language Models (MLLMs). Featuring fine-grained annotations of perceptual and reasoning complexity, the framework enables a detailed, instance-level comparison between machine and human cognitive profiles. Experiments across 14 MLLMs reveal that while human performance is predictable and shaped by abstract complexity, MLLM performance is brittle and sensitive to low-level visual cues. The work concludes that current MLLMs show early signs of spatial cognition but lack the robust, compositional understanding seen in humans.

**Strengths:**

1.  **Creation of a High-Quality Benchmark :** The paper introduces a new benchmark derived from standardized psychometric tests (11+ exams), which is a strong foundation. This ensures the tasks are well-vetted for isolating specific spatial reasoning skills and are not confounded by commonsense knowledge or linguistic ambiguity often present in other VQA datasets. The inclusion of a private test split sourced from commercial providers is a significant strength, directly addressing the critical issue of data contamination in an era where web-crawled data is likely part of MLLM training sets.

2.  **Introduction of a Fine-Grained, Multi-Feature Annotation Framework:** A strength is moving beyond simple aggregate accuracy. The paper introduces a set of cognitive annotations, including `visual perception complexity` and `general reasoning process`. This framework enables an instance-level analysis, allowing researchers to investigate *why* a model might fail on a specific problem (e.g., high perceptual load vs. complex reasoning chain). This is a conceptual leap from traditional benchmarks that only report overall scores.

3.  **Novel Cognitive-Inspired Evaluation Framework:** The paper's core proposal—to conduct a *parallel analysis of human and machine cognitive profiles*—is a powerful and novel approach. It frames the evaluation not just as measuring performance, but as comparing the underlying mechanisms and failure modes. By attempting to model what features predict success and cognitive effort for both humans and MLLMs, the paper provides a blueprint for a more explanatory and insightful form of AI evaluation.

4.  **Systematic Critique of Existing Evaluation Paradigms:** The paper provides a valuable empirical critique of the common "single composite image" evaluation method used in other benchmarks. By showing that advanced models perform significantly better when presented with separate, cropped images, it demonstrates that previous results might conflate reasoning failures with more basic visual parsing challenges. This is a crucial methodological insight for the field.

**Weaknesses:**

*   **Subjectivity in Reasoning Process Annotation:** The annotation of "General Reasoning" requires experts to decompose a thought process into a sequence of predefined atomic operations (e.g., *Pattern Matching, Spatial Manipulation*). This methodology has two key problems:
    1.  **It assumes a discrete, serial reasoning process:** Human spatial reasoning can be holistic and parallel, not necessarily a step-by-step symbolic procedure. This annotation forces a potentially artificial structure onto a fluid cognitive process.
    2.  **It is inherently subjective:** While inter-annotator agreement was high, the decomposition itself is an interpretation. Different experts might conceptualize the steps differently, and the predefined categories may not perfectly capture the nuances of human thought.

*   **Oversimplification of Visual Perception Complexity:** "Pattern complexity" is quantified by counting atomic components like lines or surfaces (lines 202-205). This objective metric ignores the Gestalt principles of perception. For example, a complex pattern with high symmetry might be perceptually easier to process than a simpler but asymmetrical one. The metric is a simplification that may not fully capture the true perceptual load on a human or a model.

*   **Limited Scope of Spatial Capabilities:** The authors select three core capabilities (Spatial Relation & Orientation, Spatial Visualization, Flexibility of Closure) from a much broader spectrum of human spatial intelligence. They justify this by excluding skills less relevant to current MLLM architectures (e.g., kinesthetic reasoning). However, this selection may still be too narrow to make sweeping claims about MLLM spatial cognition as a whole. The findings might not generalize to other important spatial tasks like navigation, mental mapping, or complex 3D assembly.

*   **Potential Overstatement of MLLM "Randomness":** The paper claims that "instance-level MLLM performance remains largely random." However, their own results show some structure. Figure 3(b) demonstrates a positive correlation between human and MLLM accuracy, suggesting performance is not entirely random across difficulty levels. Furthermore, in the predictability analysis (Figure 7), some classifiers achieve F1 scores with statistically significant p-values (e.g., Gemini 2.5 Pro's p=0.0002), indicating a performance better than chance, even if the predictive power is weak. The narrative of "randomness" might be overstated for rhetorical effect.

*   **Lack of Systematic Error Typology:** The analysis successfully identifies which *features* predict failure (e.g., pattern complexity for humans, image resolution for MLLMs). However, it does not provide a systematic analysis of the *types* of errors models make. For instance, do MLLMs consistently fail at mental rotation beyond 90 degrees? Do they confuse reflection with rotation? The qualitative examples in Figure 8 are illustrative but not comprehensive. A categorized breakdown of error types would provide more genuinely "actionable insights" for improving models.

*   **Weak Analogy Between Human Response Time and MLLM Token Count:** The paper uses human response time as a proxy for cognitive effort and compares it to the number of tokens generated by MLLMs. This analogy is tenuous. Human response time reflects complex neural processing, memory access, and serial deliberation. MLLM token count, especially in a chain-of-thought process, is more a measure of verbosity, which can be heavily influenced by prompting, fine-tuning, and architectural biases, not necessarily correlating directly with computational "effort" or reasoning depth. The finding that token count and accuracy are uncorrelated in MLLMs is interesting but may not be directly comparable to the speed-accuracy trade-off in humans.

*   **Limited Exploration of Prompting:** The paper uses simple, fixed prompts for the MLLMs (Tables 2 & 3). MLLM performance is known to be highly sensitive to prompt engineering. The methodology does not explore whether different prompting strategies (e.g., Chain-of-Thought, asking the model to verbalize its spatial transformations) could have elicited better performance. Therefore, the results may reflect the limitations of the chosen prompts as much as the models' intrinsic capabilities.

**Questions:**

Please answer the points mentioned in the weakness section

---

> ### Author Response · Authors · 2025-11-20
> **Response to Reviewer yP9B (1/3)**
>
> Thank you for your recognition of our work and your valuable suggestions. We would like to address your comments as follows to get more support:
>
> > **W1: Subjectivity in Reasoning Process Annotation**
>
> We agree that human spatial reasoning is not strictly symbolic or serial. Our annotation framework is not intended as a verbatim reconstruction of human thought, but as a consistent and interpretable **proxy** for reasoning complexity that enables AI-human comparison.
>
> To minimize bias and encourage subjectivity, we intentionally did **not** impose a target solution chain or a maximum step count. Annotators decomposed problems into predefined atomic operations (Appendix B - *General Reasoning Process*) according to their own reasoning. This ensures (1) consistent units of analysis, and (2) freedom for experts to represent diverse reasoning pathways.
>
> Our quantitative analysis shows both **high-level agreement** with Pearson Correlation of Reasoning Step Length around 0.8 and **subjective variation** as follows, confirming that our annotations are reliable but not rigidly constrained:
>
> * Mean and Variance of average reasoning-step length: 4.66 +- 1.15
>
> * Step-length exact match across annotators: 39.78%
>
> * Mean and Variance of fine-grained operation types: 4.54 +- 0.47
>
> * Fine-grained operation-type exact match: 50.48%
>
> * Strict reasoning-process exact match: 38.41%
>
> These statistics indicate that experts converge on the overall cognitive structure while retaining individual interpretations with a certain level of inherent subjectivity.
>
> > **W2: Oversimplification of Visual Perception Complexity**
>
> We intentionally used simplified perceptual metrics to ensure consistency and low ambiguity among annotators. But still, when designed, our “pattern complexity” annotation distinguishes between low-level atomic elements (lines/surfaces) and **recognizable patterns**, which is consistent with the reviewer’s point that symmetry or recognizable patterns can reduce perceptual load. For example, a square is annotated as one holistic unit, not four separate lines. We will clarify this design choice in the revision.
>
> > **W3: Limited Scope of Spatial Capabilities**
>
> We appreciate the reviewer’s concern. Our selection of Spatial Relation & Orientation, Spatial Visualization, and Flexibility of Closure follows the psychometric frameworks (e.g., the Three Stratum of Intelligence [1]) in which these abilities are recognized as foundational, orthogonal components of human spatial cognition. This choice ensures broad coverage across diverse spatial skills while keeping the evaluation tractable and theoretically grounded.
>
> Importantly, these capabilities were measured with real-world spatial aptitude tests designed by human education experts to enable controlled data collection and fair human-model comparison, and we adopt these without introducing inductive biases. Many realistic spatial tasks, such as real-world navigation or 3D assembly, embed additional environmental priors, domain-specific knowledge, or complex real-world patterns. These factors introduce more uncontrolled variance in pattern complexity and make it hard to evaluate and compare model performance with human cognition.
>
> Moreover, complex spatial behaviours such as mental mapping or navigation typically require composition of multiple capabilities (e.g. for navigation it requires spatial scanning and environment), which makes the analysis hard to tell and analyse.
>
> For these reasons, we believe our chosen capabilities provide a meaningful and theoretically motivated basis for assessing MLLMs’ spatial cognition, while broader real-world tasks remain important directions for future work.
>
> > **W4: Potential Overstatement of MLLM "Randomness**
>
> We thank the reviewer for the comment. We agree that MLLM instance-level correctness shows weak predictability, as reflected by the low F1 scores (even if statistically significantly different from random guess) in Fig. 7. To avoid misunderstanding, we will revise our wording to “weakly predictable” rather than “largely random".

---

> ### Author Response · Authors · 2025-11-20
> **Response to Reviewer yP9B (2/3)**
>
> > **W5: Lack of Systematic Error Typology**
>
> We performed a systematic qualitative error analysis (Gemini 2.5 Pro, GPT-4o with separate images) and identified four recurring failure modes:
>
> 1) **Perceptual error**: Difficulty representing fine-grained visual primitives (repetition count, small shape details or spatial structures) during reasoning. This often leads to symbolic approximations in ASCII format (e.g. X O O | X X O | ...) or with referable textual descriptions (e.g. Y-pentomino) that omit critical information.
>
> 2) **Manipulation error**: Incorrect or incomplete elicitation of the outcomes about spatial manipulations such as rotations, reflections, or multi-step spatial operations. This common failure mode is more severe when it comes to complex operations such as 3D rotations or symbol tagging, which in the end leads to inconsistent or wrong reasoning chains.
>
> 3) **Spatial relation analysis error**: High-level recognition of certain spatial relations (2D rotation and 2D translation) but failure with other spatial relations. In addition, constrained by perceptual and manipulation errors, we observe inconsistent alignment between the identified spatial relations and precise local features.
>
> 4) **Cascaded logical error**: Early perceptual or manipulation mistakes often lead models to conclude that “no option is correct,” then guess.
>
> These failure modes align with our cognitive-profile findings that low-level visual perception and manipulation are primary impactful features of model performance. We will include the qualitative analysis to the paper.
>
> > **W6: Weak Analogy Between Human Response Time and MLLM Token Count**
>
> We acknowledge that human response time and MLLM token count may measure different underlying processes. Our comparison does not aim to claim equivalence but to treat both as **approximations** for *test-time computational effort*: response time for human cognitive load, token count for model reasoning effort. We will clarify this framing and explicitly state that this analogy is approximate.

---

> ### Author Response · Authors · 2025-11-20
> **Response to Reviewer yP9B (3/3)**
>
> > **W7: Limited Exploration of Prompting**
>
> We conducted additional experiments to examine how prompt formulations influence model accuracy using three alternative prompt settings:
>
> 1) Verbal variations (Verbal Var): Reformulating the question text (“Which option image matches the correct pattern?”)
>
> 2) CoT prompting (“Think step by step and describe the spatial transformations before answering.”)
>
> 3) Hint: Prompts augmented with annotated human reasoning operation types (Appendix B).
>
> Additional experiments are all conducted with separate multiple images as inputs.
>
> *Additional Table 1. Model Performance with Different Prompting*
>
> | Model   | Original |  Verbal Var  |  CoT   |  Hint  |
> | :------ | :------: | :-----: | :----: | :----: |
> | GPT 4o  |  0.2973  |  0.3060 | 0.3049 | 0.3027 |
> |   o3    |  0.3913  |  0.3868 | 0.3945 | 0.3825 |
>
> *Additional Table 2. Predictability of Correctness with Different Prompting (\* with p < 0.01)*
>
> |                            | Original                      |                               | Verbal Var                         |                               | CoT                           |                               | Hint                          |                               |
> |----------------------------|-------------------------------|-------------------------------|-------------------------------|-------------------------------|-------------------------------|-------------------------------|-------------------------------|-------------------------------|
> | Model                      | GPT 4o                        | o3                            | GPT 4o                        | o3                            | GPT 4o                        | o3                            | GPT 4o                        | o3                            |
> | F1                         | 0.622*                        |                         0.549 |                         0.595 |                         0.585 | 0.658*                        |                         0.580 | 0.620*                        |                         0.564 |
> | AUC                        |                         0.561 |                         0.542 |                         0.462 |                         0.575 |                         0.583 |                         0.579 |                         0.569 |                         0.598 |
> | Acc                        |                         0.647 |                         0.555 |                         0.622 |                         0.600 |                         0.677 |                         0.587 |                         0.647 |                         0.572 |
>
> Across GPT-4o and o3 as representative non-reasoning and reasoning models, we observe **neglectable differences** in accuracy relative to our original prompt. Results of o3 align with the findings in OpenAI technical guide [2] that '*These (reasoning) models perform best with straightforward prompts, ..., prompting them to "think step by step" or "explain your reasoning" is unnecessary.*'
>
> Even provided with hints regarding spatial operations (Hint), the performance is not significantly different, which indicates that the core limitation is the integration of perception, manipulation, and reasoning, rather than a single isolated weakness, aligning with the error modes we identified. Further cognitive-profile analyses remain **consistent**, with low-level visual cues (specifically pattern complexity and image resolution) followed by spatial relation analysis and spatial manipulation being the most impactful features of model success.
>
> These results indicate that prompt formulation does **not** significantly affect performance, supporting the **robustness** of our findings. We will include full results in the appendix.
>
> **References**
>
> [1] Carroll, John B. "The three-stratum theory of cognitive abilities." (1997).
>
> [2] https://platform.openai.com/docs/guides/reasoning-best-practices#how-to-prompt-reasoning-models-effectively

---

### Official Review · Reviewer_qLGk · 2025-11-01

**Soundness:** 3
**Presentation:** 3
**Contribution:** 3
**Rating:** 6
**Confidence:** 4

**Summary:**

This paper introduces 11PLUS-Bench, a benchmark designed to evaluate spatial reasoning in multimodal large language models (MLLMs). Unlike existing benchmarks that rely on aggregate accuracy, 11PLUS-Bench provides instance-level cognitive feature annotations to enable predictive modeling of both human and machine responses. The benchmark includes 824 public and 91 private examples derived from realistic 11+ standardized spatial aptitude tests, annotated by domain experts for three key cognitive dimensions, Spatial Relation & Orientation (SRO), Spatial Visualization (SV), and Flexibility of Closure (FoC). Evaluation across 14 MLLMs (including GPT-4o, Gemini-2.5-Pro, and Qwen-VL-2.5) and human baselines reveals a significant gap which reveals models lag far behind human accuracy (humans ≈ 80–85%, best MLLMs ≈ 30-40%) and exhibit random, poorly predictable instance-level behavior. Analysis using SHAP values and regression models finds that human correctness is primarily driven by pattern complexity and reasoning steps, while MLLM behavior is overly influenced by low-level perceptual features such as image resolution or bounding-box layout. The paper concludes that although MLLMs show early signs of spatial cognition, their reasoning remains unstable, unstructured, and far from human-like generalization.

**Strengths:**

* Benchmark design explicitly ties to psychometric theory (Carroll’s Three-Stratum model, spatial-visualization literature), ensuring conceptual validity.
* Introduces interpretable cognitive features (pattern complexity, reasoning type, perceptual load) enabling predictive modeling rather than aggregate scoring.
* Uses response time vs. token length to measure cognitive effort and compares predictive profiles using Random Forest and SHAP analyses.
* Reports strong annotation agreement (Pearson ≈ 0.8) and statistically significant human–model correlations in difficulty perception.
* Demonstrates that MLLMs show partial emergence of spatial cognition but are overly dependent on low-level cues.
* Private test data from commercial providers ensures low contamination to facilitate ethical and reproducibility.

**Weaknesses:**

* There are only three participants and a broader demographic diversity would strengthen validity.
* ~900 items may still be modest for modern MLLM evaluation and a larger set could increase statistical reliability.
* All tasks are multiple-choice which lacks open-ended reasoning or diagram generation tasks that test richer cognitive processes.
* Some key comparisons (e.g., per-model significance tests) are summarized without full confidence intervals.

**Questions:**

1. Was there a reason as to the limit to three participants?
2. For MLLMs, do prompt formulations (e.g., “select the best option” vs. “which image matches?”) affect accuracy?
3. Can the authors classify common model failure modes (e.g., mis-segmentation vs. transformation confusion)?
4. Could fine-tuning on annotated reasoning steps yield measurable improvements in predictive alignment with human profiles?

---

> ### Author Response · Authors · 2025-11-20
> **Response to Reviewer qLGk (1/3)**
>
> Thank you for your recognition of our work and your valuable suggestions. We would like to address your comments as follows to get more support:
>
> > **W1 & Q1: There are only three participants and a broader demographic diversity would strengthen validity.**
>
> We agree that a larger and more demographically diverse participant pool would further strengthen the study. Our present design involves three expert annotators (for gold answers and cognitive features) and three participants for human evaluation. We relied on expert labour for high-quality fine-grained spatial-cognitive annotations by prioritising quality annotations over quantity. Although constrained by financial budget for human participants, our conclusions remain **robust**:
>
> 1) High cross-participant consistency: We observe strong correlations in response times across participants (Sec. 4.2 - Human Performance).
>
> 2) Stable predictive modelling: Cognitive features reliably predict human correctness across participants (Sec. 4.3).
>
> These results demonstrate **stable behavioural patterns** despite the small sample size of participants, suggesting that our conclusions are driven by **consistent cognitive signals**.
>
> > **W2: Limited size of benchmark**
>
> Our benchmark prioritizes **quality and psychometric validity over quantity**. The high-quality data is hard to collect. We crawled the raw data with 29 spatial reasoning keywords from the web. We manually **filtered** the large raw pool (from 5352 webpages and 29 hours of videos) down to 824 items that meet **strict criteria**: no ambiguity, well-defined spatial cognition problems, image quality, as well as potential copyright and privacy issues. Each item includes **fine-grained annotations** with cognitive features (bounding boxes, pattern complexity, reasoning process), giving the dataset analytical depth uncommon in MLLM benchmarks.
>
> Moreover, our thousand-level scale is comparable to widely used reasoning benchmarks (e.g., GPQA [1]: 448 items, GSM8K [2] validation set: 1319 items). In addition, our statistical analyses yield significant human–model correlations and robust predictive results for human participants, supporting the reliability of such benchmark.
>
> > **W3: All tasks are multiple-choice**
>
> The multiple-choice questions are used by **real-world spatial aptitude test**, which we keep in our work without reframing.
>
> Besides, multiple-choice design facilitates more straightforward and accurate **evaluation** in terms of correctness. This is aligned with other established reasoning benchmarks (e.g. GPQA [1], SPACE [3], etc.).
>
> In addition, multiple-choice questions also help to isolate spatial cognition from other irrelevant capabilities such as aesthetic diagram generation, which are **orthogonal** to spatial reasoning. Our controlled format allows clearer attribution of model errors to spatial cognition rather than unrelated generative abilities.
>
> > **W4: Full confidence intervals**
>
> We thank the reviewer for this suggestion. However, the permutation tests we used in our work are non-parametric significance tests that provide a p-value based on the null distribution. Therefore they do not inherently produce confidence intervals, which is standard in the statistical literature [4]. To this end, in our work, to stay consistent with common practice, we used permutation tests solely for significance with p value.

---

> ### Author Response · Authors · 2025-11-20
> **Response to Reviewer qLGk (2/3)**
>
> > **Q2: Prompt formulation**
>
> We conducted additional experiments to examine how prompt formulations influence model accuracy using three alternative prompt settings:
>
> 1) Verbal variations (Verbal Var): Reformulating the question text (“Which option image matches the correct pattern?”)
>
> 2) CoT prompting (“Think step by step and describe the spatial transformations before answering.”)
>
> 3) Hint (further elaborated in response to Q4): Prompts augmented with annotated human reasoning operation types (Appendix B).
>
> Additional experiments are all conducted with separate multiple images as inputs.
>
> *Additional Table 1. Model Performance with Different Prompting*
>
> | Model   | Original |  Verbal Var  |  CoT   |  Hint  |
> | :------ | :------: | :-----: | :----: | :----: |
> | GPT 4o  |  0.2973  |  0.3060 | 0.3049 | 0.3027 |
> |   o3    |  0.3913  |  0.3868 | 0.3945 | 0.3825 |
>
> *Additional Table 2. Predictability of Correctness with Different Prompting (\* with p < 0.01)*
>
> |                            | Original                      |                               | Verbal Var                         |                               | CoT                           |                               | Hint                          |                               |
> |----------------------------|-------------------------------|-------------------------------|-------------------------------|-------------------------------|-------------------------------|-------------------------------|-------------------------------|-------------------------------|
> | Model                      | GPT 4o                        | o3                            | GPT 4o                        | o3                            | GPT 4o                        | o3                            | GPT 4o                        | o3                            |
> | F1                         | 0.622*                        |                         0.549 |                         0.595 |                         0.585 | 0.658*                        |                         0.580 | 0.620*                        |                         0.564 |
> | AUC                        |                         0.561 |                         0.542 |                         0.462 |                         0.575 |                         0.583 |                         0.579 |                         0.569 |                         0.598 |
> | Acc                        |                         0.647 |                         0.555 |                         0.622 |                         0.600 |                         0.677 |                         0.587 |                         0.647 |                         0.572 |
>
> Across GPT-4o and o3 as representative non-reasoning and reasoning models, we observe **neglectable differences** in accuracy relative to our original prompt. Results of o3 align with the findings in OpenAI technical guide [5] that '*These (reasoning) models perform best with straightforward prompts, ..., prompting them to "think step by step" or "explain your reasoning" is unnecessary.*'
>
> Cognitive-profile analyses remain **consistent**, with low-level visual cues (specifically pattern complexity and image resolution) followed by spatial relation analysis and spatial manipulation being the most impactful features of model success.
>
> These results indicate that prompt formulation does **not** significantly affect performance, supporting the **robustness** of our findings. We will include full results in the appendix.

---

> ### Author Response · Authors · 2025-11-20
> **Response to Reviewer qLGk (3/3)**
>
> > **Q3: Common model failure modes**
>
> We performed a systematic qualitative error analysis (Gemini 2.5 Pro, GPT-4o with separate images) and identified four recurring failure modes:
>
> 1) **Perceptual error**: Difficulty representing fine-grained visual primitives (repetition count, small shape details or spatial structures) during reasoning. This often leads to symbolic approximations in ASCII format (e.g. X O O | X X O | ...) or with referable textual descriptions (e.g. Y-pentomino) that omit critical information.
>
> 2) **Manipulation error**: Incorrect or incomplete elicitation of the outcomes about spatial manipulations such as rotations, reflections, or multi-step spatial operations. This common failure mode is more severe when it comes to complex operations such as 3D rotations or symbol tagging, which in the end leads to inconsistent or wrong reasoning chains.
>
> 3) **Spatial relation analysis error**: High-level recognition of certain spatial relations (2D rotation and 2D translation) but failure with other spatial relations. In addition, constrained by perceptual and manipulation errors, we observe inconsistent alignment between the identified spatial relations and precise local features.
>
> 4) **Cascaded logical error**: Early perceptual or manipulation mistakes often lead models to conclude that “no option is correct,” then guess.
>
> These failure modes align with our cognitive-profile findings that low-level visual perception and manipulation are primary impactful features of model performance. We will add the qualitative analysis to the paper.
>
> > **Q4: Could fine-tuning on annotated reasoning steps yield measurable improvements in predictive alignment with human profiles?**
>
> We appreciate this insightful question. Although our dataset size limits full fine-tuning, we explored a lighter-weight variant as preliminary exploration: reasoning-guided prompting by injecting annotated spatial operation types into GPT-4o and o3 (Hint as in *Additional Tables* above).
>
> Across GPT-4o and o3, performance under hinted reasoning is **not** significantly different from the default condition. Because we provide the model with types of involved spatial operations, we notice that R - Spatial Manipulation and R - Spatial Relation Analysis become **less impactful** compared to other factors such as P - Image Resolution in cognitive profile analysis for GPT-4o. This indicates that the core limitation is the integration of perception, manipulation, and reasoning, rather than a single isolated weakness, aligning with the error modes we identified.
>
> We agree that full fine-tuning with specialized spatial-reasoning supervision is a promising future direction, and we leave this as future work based on our findings.
>
> **References**
>
> [1] Rein, David, et al. "Gpqa: A graduate-level google-proof q&a benchmark." First Conference on Language Modeling. 2024.
>
> [2] Cobbe, Karl, et al. "Training verifiers to solve math word problems." arXiv preprint arXiv:2110.14168 (2021).
>
> [3] Ramakrishnan, Santhosh Kumar, et al. "Does Spatial Cognition Emerge in Frontier Models?." arXiv preprint arXiv:2410.06468 (2024).
>
> [4] Phillip Good. "Permutation Tests: A Practical Guide to Resampling Methods for Testing Hypotheses". 2000.
>
> [5] https://platform.openai.com/docs/guides/reasoning-best-practices#how-to-prompt-reasoning-models-effectively

---

### Meta-Review · Area_Chair_itgw · 2026-01-05

**Summary:**

This paper proposes a benchmark for evaluating spatial reasoning in MLLMs and received mixed reviewer scores (6/6/4/4).

Reviewers (AuWm, yP9B, qLGK) raised concerns about limited scale and coverage, including a relatively small dataset, imbalanced task distribution, and incomplete model evaluation. Concerns were also expressed regarding the limited human baseline (qLGK, AuWm), the subjectivity of reasoning-step and perceptual-complexity annotations (xG19), and insufficient analysis of potential data contamination (AuWm). Additional issues include potentially overstated claims about MLLM “randomness” (yP9B) and the lack of a systematic error taxonomy (AuWm).

The authors provided detailed responses for the concerns of each reviewer.

**Reviewer Concerns:**

The reviewers’ concerns can be summarized as follows:

Reviewer qLGk: Main concerns are the small human baseline, limited dataset size, multiple-choice-only evaluation, and incomplete statistical reporting.

Reviewer yP9B: Questions the subjectivity and simplification of cognitive annotations, narrow task coverage, overstated claims of randomness, weak effort comparisons, and limited prompt/error analysis.

Reviewer AuWm: Emphasizes limited and partially closed data, lack of contamination auditing, a weak human baseline, shallow qualitative analysis, and incomplete model coverage.

Reviewer xG19: Raises concerns about ambiguous reasoning-step annotations, task imbalance, insufficient analysis of model reasoning traces, and unaddressed architectural or data confounds.

**Reviewer Scores:**

The authors provided detailed responses, and no further feedback was received from the reviewers. I think that the inherent limitations of this benchmark work, such as limited scale and coverage (small dataset, imbalanced tasks, and incomplete model coverage) and the underpowered human baseline, cannot be fully resolved within the current version of the paper, as already noted in the reviews.

---

### Decision · Program_Chairs · 2026-01-26

Reject